# Helix-bundle and C-terminal GPCR domains differentially influence GRK-specific functions and β-arrestin-mediated regulation

Edda S. F. Matthees [1,5], Raphael S. Haider [1,2,3,5], Laura Klement [1], Mona Reichel[1], Nina K. Blum[4], Verena Weitzel[1], Thimea Trüpschuch [1], Carla Ziegler[1], Julia Drube [1], Stefan Schulz [4] & Carsten Hoffmann [1] ✉

G protein-coupled receptors (GPCRs) orchestrate diverse physiological responses via signaling through G proteins, GPCR kinases (GRKs), and arrestins. While most G protein functions are well-established, the contributions of GRKs and arrestins remain incompletely understood. Here, we investigate the influence of β-arrestin-interacting GPCR domains (helix-bundle/C-terminus) on β-arrestin conformations and functions using refined biosensors and advanced cellular knockout systems. Focusing on prototypical class A (b2AR) and B (V2R) receptors and their chimeras (b2V2/V2b2), we show that most N-domain β-arrestin conformational changes are mediated by receptor C-terminus-interactions, while C-domain conformations respond to the helix-bundle or an individual combination of interaction interfaces. Moreover, we demonstrate that ERK1/2 signaling responses are governed by the GPCR helix-bundle, while β-arrestin co-internalization depends on the receptor C-terminus. However, receptor internalization is controlled via the overall GPCR configuration. Our findings elucidate how individual GPCR domains dictate downstream signaling events, shedding light on the structural basis of receptor-specific signaling.

G protein-coupled receptors (GPCRs) constitute the largest family of transmembrane receptors in humans, featuring more than 800 different members[1]. Positioned readily at the plasma membrane of cells, they sense extracellular stimuli to induce targeted intracellular signaling responses that impact many, if not all physiological processes. However, to do so, GPCRs require the activity of intracellular signal transducers, such as G proteins, GPCR kinases (GRKs) and arrestins. The availability of these proteins shapes all signaling responses of GPCRs, as they bind directly to the intracellular face of activated receptors. For G proteins and GRKs, this binding is enabled by the opening of the GPCR cytoplasmic cavity, which follows receptor activation[2,3]. Interestingly, high-affinity arrestin binding additionally relies on the phosphorylation state of GPCRs[4], which is predominantly controlled by GRKs and second messenger kinases.

While G proteins invoke primary signaling functions, such as second messenger responses or the gating of ion channels[5], GRKs and

[1]Institut für Molekulare Zellbiologie, CMB – Center for Molecular Biomedicine, Universitätsklinikum Jena, Friedrich-Schiller-Universität Jena, Jena, Germany. [2]Division of Physiology, Pharmacology and Neuroscience, School of Life Sciences, Queen's Medical Centre, University of Nottingham, Nottingham, UK. [3]Centre of Membrane Protein and Receptors, Universities of Birmingham and Nottingham, Midlands, UK. [4]Institut für Pharmakologie und Toxikologie, Universitätsklinikum Jena, Friedrich-Schiller-Universität Jena, Jena, Germany. [5]These authors contributed equally: Edda S. F. Matthees, Raphael S. Haider. ✉e-mail: carsten.hoffmann@med.uni-jena.de

arrestins have been shown to work in conjunction to promote the downregulation of GPCR signaling. Here, tight interactions with arrestins obstruct the G protein binding site[6,7], while GPCR phosphorylation and arrestin binding facilitate receptor internalization[8–10], to reduce the number of signaling-competent receptors at the plasma membrane. In recent years, however, we gathered more and more evidence, suggesting that GRKs and arrestins might also influence downstream signaling characteristics of GPCRs. In this context, one proposed function of GPCR-bound β-arrestins is the scaffolding of multi-tiered cascades of mitogen-activated protein kinases (MAPKs), such as extracellular signal-regulated kinases 1 and 2 (ERK1/2), to facilitate their sequential activation[11]. However, this function seems to be mostly supportive[12] in living cells, as GPCRs are able to activate ERK1/2 independently of β-arrestins[13,14], while G proteins are evidently required for this process[15]. Intriguingly, the human genome encodes for dozens of different combinations of G protein trimers, while only four ubiquitously expressed GRK isoforms (GRK2, 3, 5 and 6), as well as β-arrestin1 and 2 fulfill targeted functions for most nonvisual GPCRs.

This is especially interesting, as different GPCRs can exhibit distinct interaction modes with β-arrestins, which result in diverging downregulation processes[16]. The discovery of these characteristics even merited a separate classification system for GPCRs[17], as class A receptors (e.g. the β2 adrenergic receptor (b2AR)) show transient β-arrestin interactions and a higher affinity for β-arrestin2, while class B receptors (e.g. the arginine-vasopressin receptor type 2 (V2R)) show longer-lived interactions with both β-arrestin isoforms and internalize together in complex with them. This suggests that GPCRs encode for these different behaviors, to inform GRKs and β-arrestins and enable them to carry out receptor-specific functions. While it is clear that this information is conveyed via the specific complex formation between different receptors and β-arrestins, we do not know which of the involved GPCR domains orchestrate individual GRK- and β-arrestin-mediated functions.

Certainly, numerous studies already investigated these mechanisms, specifically focusing on the phosphorylated GPCR C-terminus. Via the utilization of synthesized peptides that mimic these receptor C-termini, it was shown that different phosphorylation patterns induce differential arrestin conformational changes, which, in turn, would support specific functions[18,19]. However, current research either completely neglects the influence of the GPCR transmembrane helix-bundle[18–21], or does not exhibit the required resolution to detect conformational changes in individual β-arrestin domains[22]. Moreover, even though structural biologists increasingly succeed in the elucidation of GPCR–β-arrestin complex structures, the interpretation of results is often complicated due to critical receptor modifications, required to stabilize these complexes[23,24]. Hence, neither of these approaches were effective in delineating the individual contributions of the GPCR transmembrane helix-bundle and phosphorylated C-terminus to individual β-arrestin conformational states and functions.

Using a refined set of β-arrestin conformational biosensors, we recently uncovered β-arrestin conformational fingerprints that are induced by ligand-activation of the presumably unphosphorylated parathyroid hormone 1 receptor (PTH1R), in the absence of GRKs[25]. This confirms that both binding interfaces, namely the GPCR transmembrane helix-bundle and C-terminus, influence the resulting active β-arrestin conformation.

In this study, we aim to characterize these differential influences and further investigate how they induce targeted β-arrestin functions. Hence, we decided to revisit two extensively studied, prototypical class A and B GPCRs: the b2AR and V2R. Furthermore, we focus on the individual impact of their transmembrane helix-bundles and C-termini via the utilization of chimeric receptors that feature switched C-terminal domains, termed b2V2 and V2b2[16]. With this, we analyze receptor variant-specific β-arrestin interactions and concomitant β-arrestin conformational change signatures as well as their proposed downstream consequences in the form of receptor internalization and ERK1/2 phosphorylation. Furthermore, using recently generated GRK knockout cells[26], we assess these processes in a GRK-dependent manner. Leveraging this unique biophysical and cell biological toolkit, we are now able to attribute conformational changes in specific β-arrestin domains, as well as their proposed functions as being caused by either the GPCR transmembrane helix-bundle, the GPCR C-terminus or a distinct combination of these domains.

## Results

### Model class A and B GPCRs induce distinct patterns of β-arrestin conformational changes

To assess differences in β-arrestin interactions for class A and B receptors, we focused on two well-established representatives, b2AR and V2R, respectively[17]. Via intermolecular BRET (Fig. 1a), we measured β-arrestin recruitment to the b2AR (Fig. 1b and Supplementary Fig. 1a, e, f) and V2R (Fig. 1c and Supplementary Fig. 1c, g, h). The time-dependent data show that both receptors form stable interactions with β-arrestin2, reaching a plateau two minutes after ligand addition. We utilized a recently published set of biosensors[25] to measure activation-dependent β-arrestin conformational change fingerprints resulting from the complex formation with the GPCRs (Fig. 1d). As an example, the intramolecular BRET signal for β-arrestin conformational changes, measured with the β-arrestin2-F5 biosensor, are shown in Fig. 1e, f and Supplementary Fig. 1b, d (β-arrestin1 in Supplementary Fig. 1i–l). For β-arrestin2, these results mirror the recruitment data and display stable molecular rearrangements for both receptors over time (Fig. 1e, f). Notably, we were not able to assess β-arrestin1 conformational changes induced by the b2AR (Supplementary Fig. 1i, j), therefore, we omitted the analysis of the b2AR–β-arrestin1 interaction from this study.

Complete β-arrestin2 conformational change fingerprints for the interaction with b2AR and V2R are shown in Fig. 1g, h (V2R–β-arrestin1 in Supplementary Fig. 1m). Additionally, we plotted the β-arrestin2 conformational changes onto its surface structure to enable visual attribution of molecular rearrangements (Fig. 1i, j, V2R–β-arrestin1 in Supplementary Fig. 1n). As the foundation of these datasets, all β-arrestin conformational change measurements were subjected to thorough pharmacological analysis, employing a four-parameter fit for all β-arrestin change data (Supplementary Fig. 2). According to our analysis, we conclude that data from individual biosensors failing to be fitted or to meet criteria regarding the Hill slope (absolute value > 0.1) and $EC_{50}$ (between $10^{-3}$ and $10^{0.3}$ μM) values did not exhibit measurable conformational changes. Such receptor–β-arrestin biosensor conditions were classified as pharmacologically non-responding and assigned a value of zero (Supplementary Fig. 2).

The b2AR only induced measurable β-arrestin2 conformational changes in the phosphorylation-sensing N-domain, specifically at positions F2, F3 and F5. In contrast, we were able to record V2R-induced β-arrestin2 conformational change signals for all N-domain-localized biosensors, as well as F10 and F1 in the C-domain (Fig. 1g–j).

The phosphorylation-sensing N-domain rearrangements indicate a more pronounced interaction of β-arrestin2 with the V2R C-terminus compared to the b2AR, while differences in C-domain conformational changes, specifically F1, might suggest distinct membrane association of the β-arrestin2 C-edge loops when coupling to b2AR and V2R.

We further analyzed GRK-dependent GPCR phosphorylation and β-arrestin interactions of b2AR and V2R. To identify the influence of individual β-arrestin-interacting GPCR domains, specifically the transmembrane helix-bundle and C-terminus, we additionally characterized the chimeric V2b2 and b2V2 receptors.

### Different combinations of GPCR helix-bundles and C-termini orchestrate distinct, phosphorylation-dependent β-arrestin interactions

A schematic representation of analyzed b2AR- and V2R-based receptor variants is shown in Fig. 2a. To construct the chimeric V2b2 and b2V2

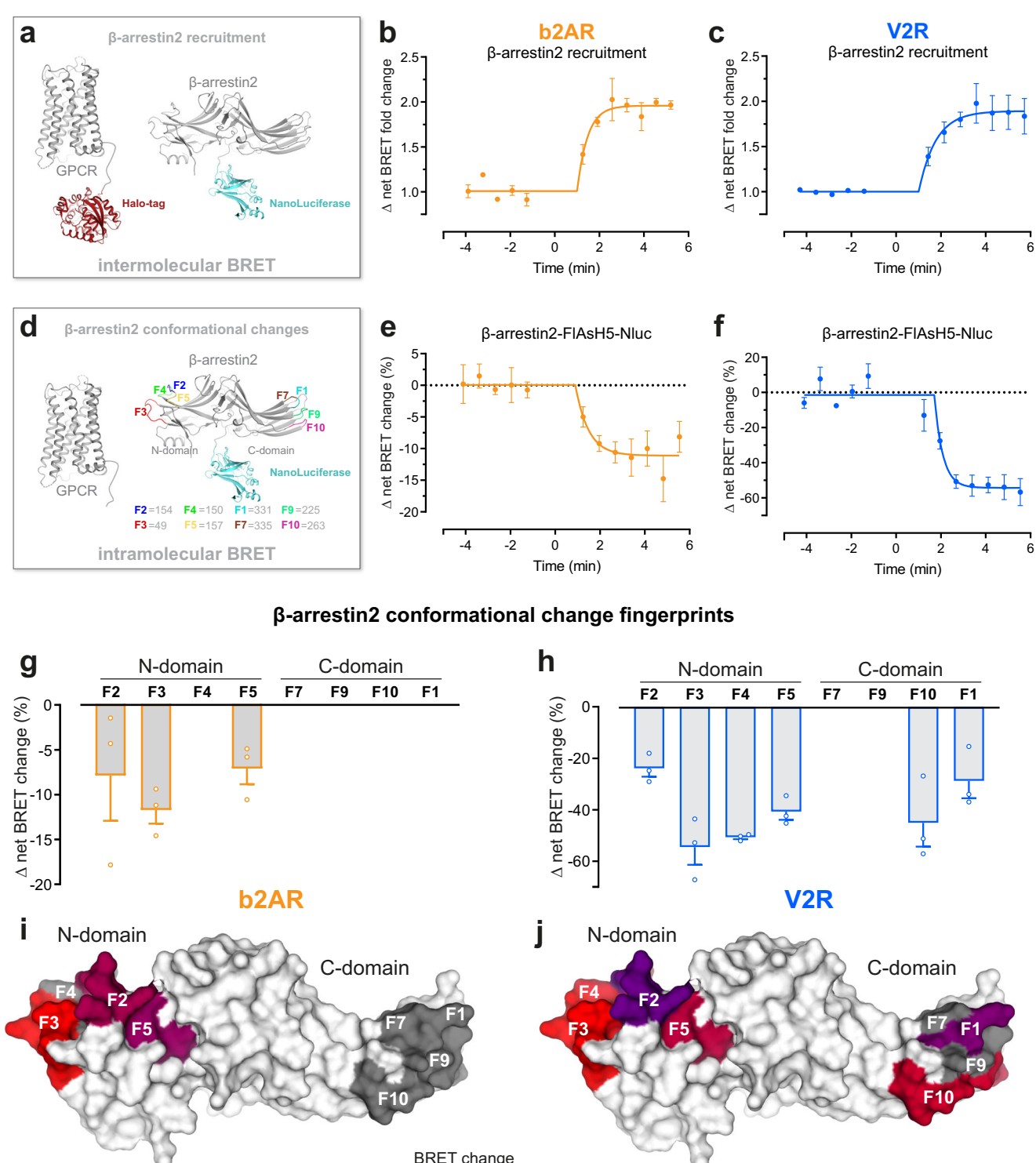

**β-arrestin2 conformational change fingerprints**

receptors, we exchanged the respective C-termini, similar to Oakley et al.[16]. For the investigation of GRK-dependent receptor phosphorylation, we focused on two phosphorylation clusters per C-terminus, proximal and distal, respectively (Fig. 2b). Specifically, for the V2R C-terminus, multiple studies identified charge-charge interactions between the two phosphorylation clusters (T359, T360 and S362, S363, S364) and the β-arrestin N-domain[21,27]. Similarly, mutational studies focusing on the b2AR C-terminus confirm that the two respective clusters (S355, S356 and T360, S364) are important for interactions with β-arrestins[28,29].

To allow for quantification of obtained results, we created cell lines (previously described HEK293 Control cells and quadruple GRK2,3,5,6 knockout cell line (ΔQ-GRK)[26]), stably overexpressing one of the four HA-tagged receptor variants and sorted them for similar expression levels via fluorescence-activated cell sorting (FACS). Using a bead-based 96-well assay[30] in combination with these stable cell lines, we assessed the GRK-dependency of proximal and distal receptor phosphorylation (Fig. 2c–j).

Our data clearly show that both clusters in the two receptor C-termini are phosphorylated in an agonist-dependent manner.

**Fig. 1 | Model class A and B GPCRs, β2 adrenergic receptor (b2AR) and Vasopressin receptor 2 (V2R), induce distinct patterns of β-arrestin conformational patterns. a** Schematic representation of NanoBRET-based assay to measure β-arrestin2-NanoLuciferase (Nluc) recruitment to the GPCR-Halo-Tag in Control cells. For the b2AR measurements, the BRET pair was switched. **b, c** β-arrestin2 recruitment to b2AR (**b**) and V2R (**c**), as published in Drube et al.[26]. Here, the Δ net BRET fold change was analyzed over time when cells were stimulated with 10 μM Isoprenaline (Iso) (**b**) or 3 μM [Arg8]-Vasopressin (AVP) (**c**). Timepoint 0 is defined as midpoint of the average time needed for ligand addition, considering all analyzed biological and technical replicates. All data are shown ±SEM of $n = 3$ independent experiments. **d** Schematic visualization of the utilized intramolecular NanoBRET-based β-arrestin2 sensors, as described in detail in Haider et al.[25]. **e, f** Conformational change of β-arrestin2-FlAsH (F) 5-Nluc in Control cells, co-

transfected with untagged b2AR (**e**)[26] or V2R (**f**). Data were analyzed over time as in (**b, c**) and are shown as Δ net BRET change in percent. **g–j** Fingerprint of β-arrestin2 conformational change sensors measured in Control cells in presence of untagged b2AR (**g, i**) or V2R (**h, j**). The Δ net BRET change at 10 μM Iso (**g**) or 3 μM AVP (**h**) are shown as bar graphs. All data are shown ±SEM of $n = 3$ independent experiments. Sensor conditions, which did not fulfill the set pharmacological parameters (Hill slope and $EC_{50}$ analysis as described in methods) were classified as non-responding conditions and assigned zero. These data were visualized onto the surface of the inactive β-arrestin2 crystal structure (PDB: 3P2D) by coloring the respective loop (-fragments) of the labeled FlAsH positions (**i, j**) ranging from blue to red. For each receptor, the Δ net BRET change was normalized to the maximally reacting sensor (red). Non-responding conditions are shown in gray. Source data are provided as a Source Data file.

---

Furthermore, this phosphorylation is abolished in the absence of GRKs, confirming their essential function in this process (Fig. 2c–j).

Similarly, β-arrestin2 recruitment to the tested GPCR variants is dependent on the availability of GRKs, albeit to different extents (Fig. 2k–n, β-arrestin1 in Supplementary Fig. 3a–d). Nevertheless, we were also able to monitor a residual GRK-independent β-arrestin2 recruitment to all receptor variants, except the V2b2. Here, the differences between V2R and V2b2 possibly indicate that the geometry of utilized binding interfaces is less favorable in case of the V2R-based chimera (Fig. 2l, n). This contrasts with our initial expectation that the GRK-independent β-arrestin recruitment would be primarily mediated by the transmembrane helix-bundle.

Following this, we investigated the GRK-dependency of the complete β-arrestin conformational change fingerprints, induced by the four receptor variants, analogously to Fig. 1g, h. The β-arrestin2-F5 biosensor shows similar conformational change signals for interactions with receptors featuring the b2AR C-terminus, regardless of GRK presence (Fig. 2o, p). In contrast, β-arrestin2-F5 signals are drastically reduced for β-arrestin interactions facilitated by the V2R C-terminus in the absence of GRKs (Fig. 2q, r). Nevertheless, the V2R still induces the largest GRK-promoted β-arrestin molecular rearrangements when measured with β-arrestin2-F5, in comparison to the other three receptor variants. To enable the comparison of complete GRK-dependent β-arrestin2 conformational change fingerprints we plotted the data for all sensor positions as radar charts (Fig. 2s–v, β-arrestin1 in Supplementary Fig. 3e–g). Here, exchange of the GPCR C-terminus mostly influences β-arrestin2 conformational changes in the N-domain, yielding a characteristic pattern shared between b2V2 and V2R (Fig. 2u, v). This was expected, since the β-arrestin N-domain is the main interaction interface for phosphorylated GPCR C-termini[31,32]. Additionally, β-arrestin conformational changes induced by receptors containing the V2R C-terminus show a pronounced GRK-dependency. The b2AR C-terminus influences the β-arrestin2 N-domain conformational changes in a characteristic manner as well. Yet, measured molecular movements at position F3 were abolished when interacting with V2b2 as opposed to the interaction with b2AR (Fig. 2s, t).

We also found striking differences between β-arrestin2 conformational changes induced by the different transmembrane helix-bundles. Our data suggest that C-domain conformational changes, specifically in the F10 position, are exclusively mediated in presence of receptors featuring the V2R helix-bundle in a GRK-dependent manner. Moreover, the loss of β-arrestin2-F10 signal in the absence of GRKs is shared between interactions with the V2b2 and V2R (Fig. 2t, v). In contrast, when stimulating receptors containing the b2AR transmembrane helix-bundle in Control cells, no β-arrestin2 conformational changes are measurable in its C-domain (Fig. 2s, u).

In summary, our experiments show that the receptor C-termini are able to induce characteristic conformational changes in the β-arrestin N-domain, while C-domain conformational changes are mostly influenced by the specific GPCR helix-bundle.

## The kinetic profile of ERK1/2 phosphorylation is governed by the GPCR transmembrane helix-bundle

One described downstream function of β-arrestins is the scaffolding of MAPKs[33]. To characterize the influence of the two assessed GPCR–β-arrestin binding interfaces on this, we investigated the activation of ERK1/2 by the four receptor variants.

Grundmann et al. showed that ERK1/2 phosphorylation is strictly dependent on G protein activity[15]. Nevertheless, various studies report on a modulating function of β-arrestins regarding the amplification of MAPK signaling[34,35]. Since we found striking differences in β-arrestin interactions with the different receptor variants, we wanted to specifically characterize the GRK-dependent influence of GPCR C-termini and helix-bundles on ERK1/2 activation. Hence, we analyzed ERK1/2 phosphorylation in Control and ΔQ-GRK cells, stably overexpressing one of the receptor constructs, via Western blot and up to 30 min after agonist stimulation (Fig. 3). To ensure comparability between Control and ΔQ-GRK cells, the cells were sorted for similar receptor expression for each transmembrane helix-bundle.

Our data show striking differences in the time-dependent profiles of ERK1/2 activation between the utilized receptor transmembrane helix-bundles. Typical for class A GPCRs, activation of the b2AR induces a transient ERK1/2 phosphorylation, peaking at two minutes and returning to basal levels after ten minutes (Fig. 3a, e). In contrast, the V2R, as a class B receptor[34,36], shows a prolonged ERK1/2 activity profile, reaching its maximum at the five minutes time point with a more gradual signal decay (Fig. 3d, h). The chimeric b2V2 displays a transient ERK1/2 response similar to the b2AR (Fig. 3c, g), while the kinetic ERK1/2 signaling profile of the V2b2 resembles the V2R results (Fig. 3b, f). These experiments clearly indicate that the kinetic profile of ERK1/2 phosphorylation is mediated by the GPCR helix-bundle.

Interestingly, from these experiments we are not able to conclude whether the presence of GRKs enhances or diminishes GPCR-induced ERK1/2 signaling responses. Moreover, the influence of GRKs on these processes seems to be receptor-specific, as only the b2AR shows a reduction in ERK1/2 phosphorylation in the absence of GRKs.

To assure that the expression of endogenous β-arrestins was not limiting their capacity to scaffold MAPKs in complex with the stably overexpressed GPCRs, we additionally conducted experiments using transient β-arrestin1 and 2 overexpression conditions (Supplementary Fig. 4). These experiments yielded very similar time courses of ERK1/2 activation as compared to the conditions featuring endogenous β-arrestin expression (Supplementary Table 1). Here, we were not able to find clear mechanistic evidence for either supporting or desensitizing functions of regulatory GRKs and β-arrestins regarding GPCR-mediated ERK1/2 signaling.

We conclude from our data that ERK1/2 activity is predominantly mediated by the receptor helix-bundle. If we assume this would be a result of GPCR-specific G protein coupling, this would align well with the observations by Grundmann et al.[15] and the notion of it being a G protein-initiated mechanism, however these analyses also highlight

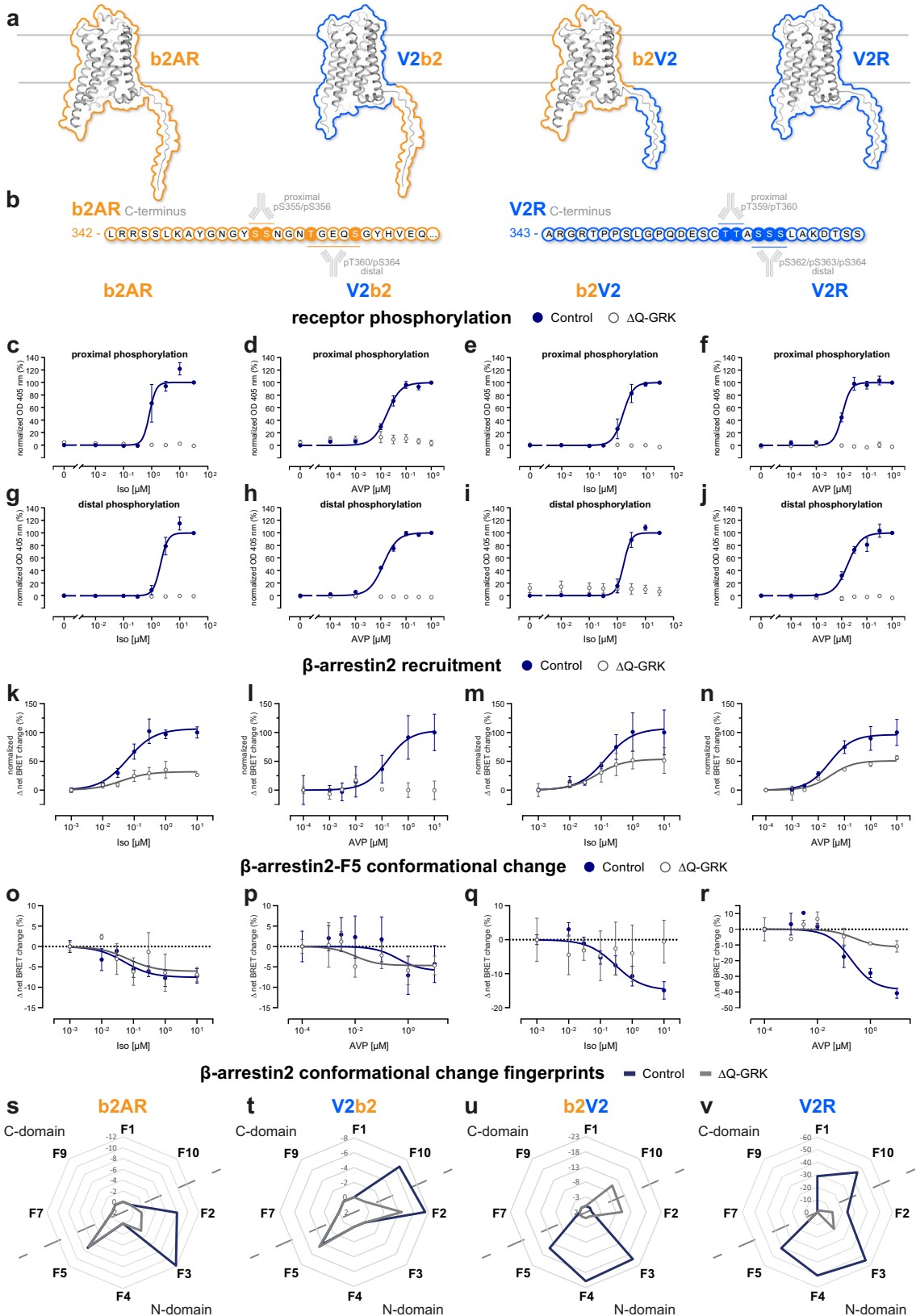

that more detailed research is required to uncover how GRKs and β-arrestins influence GPCR-induced MAPK signaling in living cells.

Since we observed various GRK-dependent differences regarding β-arrestin interactions of the four receptor variants, we further investigated whether different individual GRK isoforms facilitate distinct or similar functions.

**Differential GPCR phosphorylation by cytosolic and membrane-bound GRK isoforms induces distinct β-arrestin interactions with (chimeric) class A and B GPCRs**

Ubiquitously expressed GRK isoforms, namely GRK2, 3, 5 and 6 are grouped into two families. GRK2 and 3 are localized in the cytosol and require G protein activation to be efficiently recruited to active

**Fig. 2 | Different combinations of GPCR helix bundles and C-termini orchestrate distinct, GRK-dependent β-arrestin2 interactions. a** Schematic representation of the utilized receptor constructs and indication of the exchanged amino acids to create the chimeric constructs (b2V2 with the receptor transmembrane helix bundle of the b2AR and C-terminus of the V2R and vice versa for the V2b2) after Oakley et al.[16]. **b** (Partial) amino acid sequence of the b2AR and V2R C-terminus, including a schematic representation of the phospho-sites, targeted with specific antibodies. Antibody shape created in BioRender. Klement, L. (2025) https://BioRender.com/721vno0. **c–j** Measurements of proximal and distal C-terminal phosphorylation of b2AR (**c, g**), V2b2 (**d, h**), b2V2 (**e, i**) and V2R (**f, j**) in quadruple GRK2/3/5/6 knockout cells (ΔQ-GRK) or Control cells, utilizing a bead-based GPCR phosphorylation immunoassay[30]. The stably expressed b2AR (**c, g**) and b2V2 (**e, i**) were stimulated with different concentrations of Iso and the V2b2 (**d, h**) and V2R (**f, j**) with AVP, as indicated. Data are shown as optical density (OD) at 405 nm ±SEM of *n* = 5 independent experiments, normalized to the maximum ligand concentration for each receptor in Control cells, respectively. **k–n** β-

arrestin2 recruitment to the b2AR (**k**), V2b2 (**l**), b2V2 (**m**) and V2R (**n**) in ΔQ-GRK or Control cells stimulated with Iso (**k, m**) or AVP (**l, n**) as indicated. Data for b2AR, b2V2 and V2R (**k, m, n**) were published in Drube et al.[26] and are shown again to allow a direct comparison. For each receptor, Δ net BRET changes ±SEM of *n* = 3 independent experiments measured in ΔQ-GRK cells are shown in percent, normalized to the respective maximal change in Control cells. **o–r** β-arrestin2-F5 conformational changes in presence (Control) or absence of endogenous GRK2/3/5/6 (ΔQ-GRK) when coupling to the b2AR (**o**), V2b2 (**p**), b2V2 (**q**) or V2R (**r**). Data of *n* = 3−5 independent experiments are shown as Δ net BRET change (%) ± SEM (exact n numbers for each receptor-, GRK- and sensor-specific condition can be accessed in the source data). **s–v** Complete β-arrestin2 conformational change fingerprints are shown for each receptor in Control and ΔQ-GRK cells, stimulated with 10 μM Iso (**s, u**) or 3 μM AVP (**t, v**), as Δ net BRET change (%) in radar plots. Sensor conditions, which did not fulfill the pharmacological parameters (Hill slope and EC$_{50}$ analysis as described in methods) were classified as non-responding conditions and assigned zero. Source data are provided as a Source Data file.

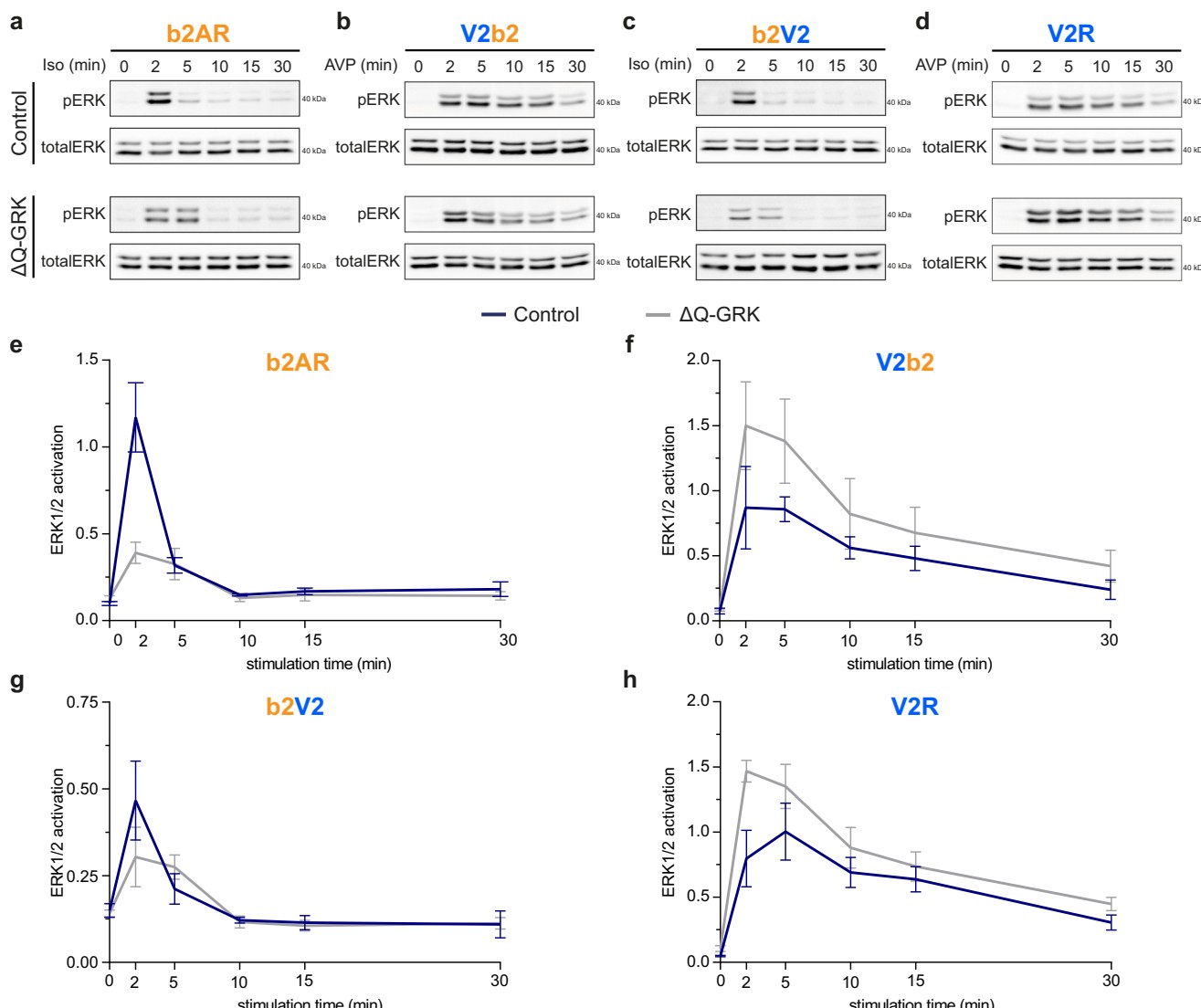

**Fig. 3 | The kinetic profile of ERK1/2 phosphorylation is governed by the GPCR transmembrane helix bundle. a–d** Representative Western blots of phosphorylated ERK (pERK) and total ERK levels over time in Control or ΔQ-GRK cells, stably expressing the b2AR (**a**), V2b2 (**b**), b2V2 (**c**) or V2R (**d**). Cells were stimulated with 1 μM Iso (**a, c**) or 100 nM AVP (**b, d**) for the time indicated. **e–h** Data of *n* = 3

independent experiments shown in (**a–d**) were quantified and are shown as ERK1/2 activation over time ±SEM (pERK1/2 divided by total ERK1/2). To compare the pERK changes over time, the area under the curve (AUC) was quantified for each condition (Supplementary Fig. 4) and complete results of the statistical analysis can be accessed in Supplementary Table. 1. Source data are provided as a Source Data file.

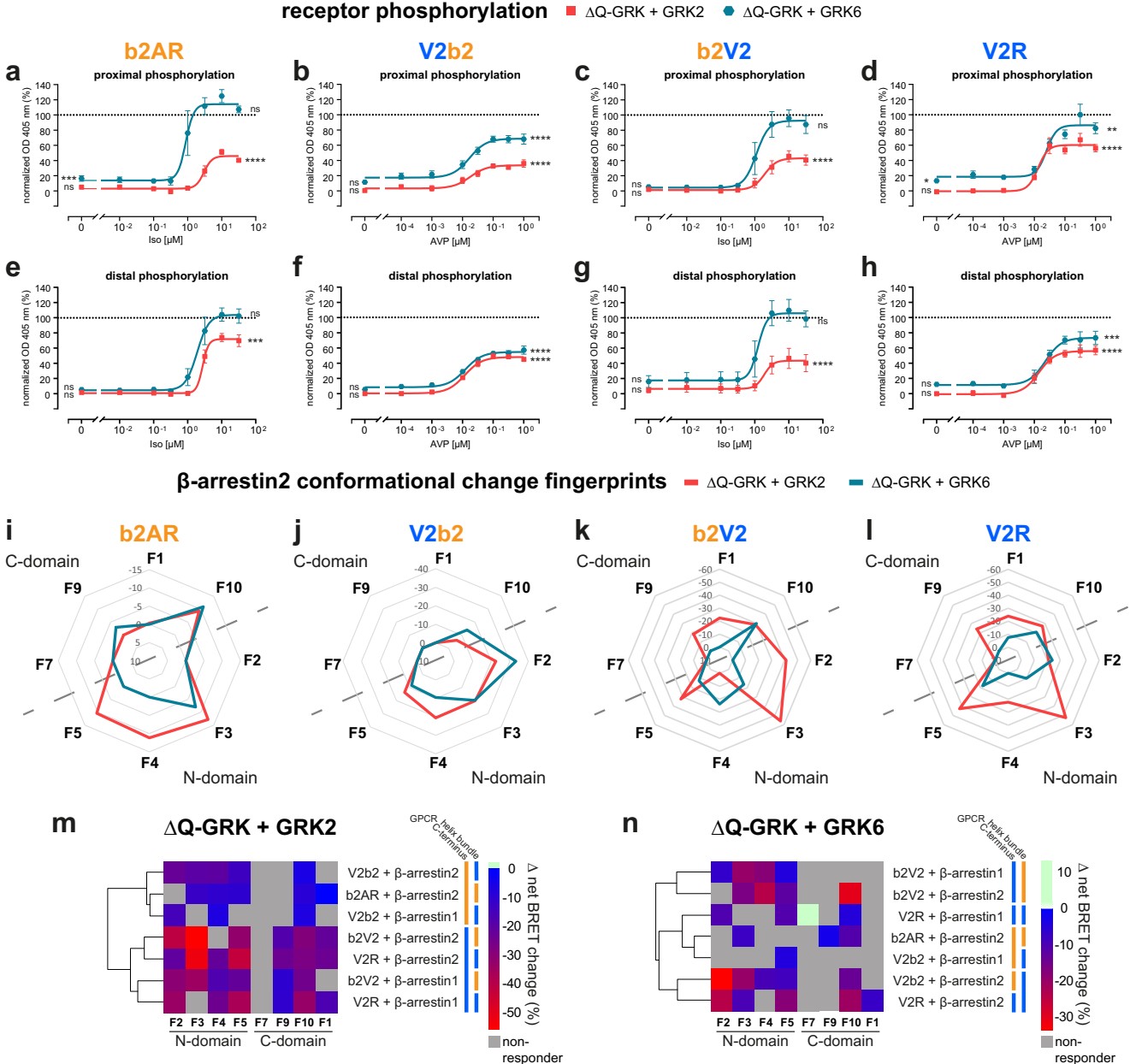

**Fig. 4 | Differential GPCR phosphorylation by cytosolic and membrane-bound GRK isoforms induces distinct β-arrestin interactions with (chimeric) class A and B GPCRs. a–h** Measurements of proximal and distal C-terminal phosphorylation in ΔQ-GRK stably expressing b2AR (**a**, **e**), V2b2 (**b**, **f**), b2V2 (**c**, **g**) or V2R (**d**, **h**) and GRK2-YFP or GRK6-YFP, analogously to Fig. 2c–j. Data are shown as optical density (OD) at 405 nm ±SEM of $n = 5$ independent experiments, normalized to the maximum ligand concentration for each receptor in Control cells, respectively. Statistical differences between measurements in ΔQ-GRK + GRK2, ΔQ-GRK + GRK6 or Control cells were compared using a two-way analysis of variance (ANOVA), followed by a Tukey's test for vehicle or highest stimulating concentrations (\*$p < 0.05$; \*\*$p < 0.01$; \*\*\*$p < 0.001$; \*\*\*\*$p < 0.0001$; ns not significant). Complete results of the statistical analysis can be accessed in Supplementary Table 2.

**i–l** Fingerprint of β-arrestin2 conformational change sensors measured in ΔQ-GRK cells, individually overexpressing GRK2 or GRK6, in presence of untagged b2AR (**i**), V2b2 (**j**), b2V2 (**k**) or V2R (**l**). The Δ net BRET changes at 10 μM Iso (**i**, **k**) or 3 μM AVP (**j**, **l**) are shown as Δ net BRET change (%) in radar plots, analogously to Fig. 2s–v. Sensor conditions, which did not fulfill the set pharmacological parameters (Hill slope and EC50 analysis as described in methods) were classified as non-responding conditions and assigned zero. **m**, **n** The Δ net BRET changes (%) at highest stimulating ligand concentrations for each FlAsH position were clustered for all receptor–β-arrestin pairs in presence of GRK2 (**m**) or GRK6 (**n**) according to Manhattan distance. GPCR transmembrane helix bundle or C-terminus identity is indicated by colored bar (b2AR in orange, V2R in blue). Source data are provided as a Source Data file.

GPCRs[37,38]. In contrast, GRK5 and 6 are present at the plasma membrane due to post-translational fatty acid conjugation[39–41].

To cover both GRK families in this study, we further focused on GRK2 and 6 as representative isoforms. While we expect that GRKs of the same family behave similarly to each other, it is possible that the untested GRK3 and GRK5 variants elicit diverging functions, not covered in this study. Analogous to our investigation of general GRK-

dependency of GPCR phosphorylation in Fig. 2, we analyzed GRK2- and 6-specific proximal and distal phosphorylation of the four receptor variants (Fig. 4a–h). To achieve this, we stably overexpressed GRK2- or 6-YFP in ΔQ-GRK cells stably overexpressing each of the GPCR variants, respectively, adding eight more stable transfectants, leading to a total number of sixteen different cell lines used in this study. Subsequently, we sorted the cells for similar YFP fluorescence via FACS to ensure that

the kinases show comparable expression. The GRK2- and 6-specific proximal or distal receptor phosphorylation data obtained in these cell lines are shown as concentration-response curves in Fig. 4a–h, normalized to maximal OD 405 values measured in Control cells (Fig. 2c–j).

Here, we observed that GRK6 was generally more efficient in mediating the phosphorylation of assessed proximal and distal clusters in comparison to GRK2. In the case of distal V2b2 phosphorylation, GRK2 and GRK6 induced most similar responses (Fig. 4f).

Notably, for receptors featuring the b2AR helix-bundle, the individual overexpression of GRK6 mediates receptor phosphorylation at 100% of Control levels for both clusters (Fig. 4a, c, e, g). This is not the case for receptors that feature the V2R transmembrane domain (Fig. 4b, d, f, h). Interestingly, GRK6 overexpression-specific basal levels of proximal receptor phosphorylation are significantly increased for b2AR and V2R (Fig. 4a, d, Supplementary Table 2). A tendency was also seen for the GRK6-specific proximal V2b2 phosphorylation and the distal V2R phosphorylation (Fig. 4b, h). In contrast, GRK2-mediated phosphorylation is not increased at baseline (vehicle-stimulated) compared to in Control cells for any of the investigated receptors.

In contrast to our expectations, our results suggest that GRK-specific GPCR phosphorylation is not strictly mediated by the C-terminal GPCR sequence. These data indicate that efficient C-terminal receptor phosphorylation requires the formation of a GPCR–GRK complex, which is likely mediated by the transmembrane helix-bundle[3,42].

Next, we investigated the consequences of GRK-specific GPCR phosphorylation on β-arrestin conformational changes induced by the four receptor variants (Fig. 4i–l, β-arrestin1 in Supplementary Fig. 5). Compared with β-arrestin conformational changes facilitated by endogenous GRK expression levels (Fig. 2s–v), GRK overexpression generally yields higher intramolecular BRET values for all receptor-specific conditions. This increase in signal amplitude upon GRK2 or 6 overexpression is most pronounced for the chimeric V2b2 and b2V2 receptors (Fig. 4j, k). Driving the system via GRK overexpression, we now also monitored pronounced β-arrestin2 conformational changes in the C-domain for the interaction with b2AR (Fig. 4i), which was not the case under endogenous GRK expression levels (Fig. 1g).

Comparing the two kinase families, our data show differences for β-arrestin2 conformational changes between GRK2- and 6-specific receptor phosphorylation for each of the four variants. These differences are more pronounced for receptor variants that feature the V2R C-terminus (Fig. 4k, l). In contrast to the relatively lower receptor phosphorylation mediated by GRK2 in comparison to GRK6 (Fig. 4a–h), the measured conformational change signals in presence of GRK2 were higher for all but four β-arrestin2 sensor positions (β-arrestin2-F9: b2AR; β-arrestin2-F2, -F10: V2b2; β-arrestin2-F4: b2V2). This might suggest that GRK2 plays additional roles in the formation of GPCR–β-arrestin complexes, other than receptor phosphorylation[43].

Conformational change patterns in the β-arrestin2 N-domain measured for the GRK2-specific interaction with b2AR and V2b2 are similar, except for the F2 sensor position (Fig. 4i, j). This is also the case for GRK2-specific V2R C-terminus-induced β-arrestin2 conformational changes, yet these show an additional difference at the F4 position (Fig. 4k, l). In order to delineate emerging patterns in GRK-dependent β-arrestin conformational change fingerprints, we clustered the Δ net BRET changes (%) at highest stimulating ligand concentrations of each FlAsH position for all measured receptor–β-arrestin pairs in presence of GRK2 (Fig. 4m) or GRK6 (Fig. 4n) according to their similarity. In presence of GRK2, receptor-specific β-arrestin conformational changes are distinctly clustered based on the GPCR C-terminus (also highlighted by the colored bars next to the clustering heatmap). This indicates that GRK2-facilitated β-arrestin conformational change fingerprints exhibit the highest similarities when interacting with receptors featuring the same C-terminus. Therefore, we conclude that GRK2-

dependent β-arrestin fingerprints are primarily dictated by the available GPCR C-terminus (Fig. 4m). However, this pattern is not observed in presence of GRK6 (Fig. 4n). Of note, the GRK6-specific conformational change fingerprints induced by receptors featuring the V2R transmembrane helix-bundle cluster together for β-arrestin2. This clustering demonstrates high similarities when β-arrestin2 interacts with V2R or V2b2, while β-arrestin1 and 2 present distinctly different fingerprints. This might indicate that GRK6-induced β-arrestin2 conformational changes are most prominently influenced by the available GPCR transmembrane domain. Yet, this characteristic seems to be receptor-specific, as it is not observed for b2AR and b2V2 (Fig. 4n).

β-arrestin conformations are clearly a result of the unique receptor complex geometry, influenced by the GPCR primary sequence, its activation state and GRK-specific phosphorylation. In fact, our data show different β-arrestin conformational changes in presence of GRK2 or 6 for all receptor variants. This behavior might result from differential GPCR phosphorylation, yet it is not the only decisive parameter. Globally, we were not able to attribute complete β-arrestin conformational change signatures to individual GPCR domains or GRK functions. However, our data show a trend indicating that GRK2-induced β-arrestin conformational changes appear to be particularly influenced by the GPCR C-terminus. In presence of GRK6, we monitored similar molecular rearrangements within β-arrestin2 when interacting with receptors featuring the V2R transmembrane helix-bundle, suggesting a receptor- and β-arrestin isoform-specific observation. This further confirms that β-arrestins are able to flexibly adopt different active confirmations for every possible GPCR–β-arrestin complex.

**β-arrestin delivery to early endosomes is controlled by the receptor C-terminus.** Given the differences we found in the GRK-dependent ERK1/2 signaling response of the four receptor variants, we further aimed to explore potential GRK-specific differences in the early trafficking of different GPCR–β-arrestin complexes. To address this, we employed confocal microscopy, utilizing Control or ΔQ-GRK cells transfected with GPCR-CFP, β-arrestin1- or 2-YFP and Rab5-mCherry as an early endosome marker. Additionally, GRK2- and 6-specific effects were investigated by overexpression in ΔQ-GRK cells. We collected at least 28 images per condition before and after ligand application and quantified the co-localization between β-arrestin and Rab5, as well as the co-localization between the receptor variants with Rab5 to assess the delivery of these proteins to early endosomes upon receptor stimulation. The images were segmented and analyzed via Squassh and SquasshAnalyst (Supplementary Fig. 6)[44,45].

Activation of all four receptor variants induces β-arrestin translocation to the plasma membrane in Control cells after 15 min of agonist stimulation, as well as receptor co-localization with Rab5 (Fig. 5a, Supplementary Fig. 7). Additionally, we also recorded at least three independent movies for each receptor following its translocation to early endosomes over the course of 15 min in Control and ΔQ-GRK cells (Supplementary Fig. 8, representative movies for all receptor variants in Control cells can be accessed via Supplementary Movies 1–4). While the temporal resolution of these experiments makes it difficult to draw detailed conclusions about the internalization kinetics, we observed a time-dependent increase in receptor co-localization with Rab5 when GRKs are present (Supplementary Fig. 8). In line with previous research[16], only the receptors featuring the V2R C-terminus, b2V2 and V2R, were able to mediate a significant increase in co-localization between β-arrestin and Rab5, compared to basal values in the respective empty vector (EV)-transfected ΔQ-GRK condition (Fig. 5a, b, Supplementary Fig. 9, Supplementary Table 3). To contextualize all measured conditions, we displayed the assessed Rab5-co-localization for receptor variants (horizontal axis) and β-arrestin2 (vertical axis) as 2-dimensional scatter plots, normalized to the highest values featured in the displayed dataset (Fig. 5c–f, β-arrestin1 in

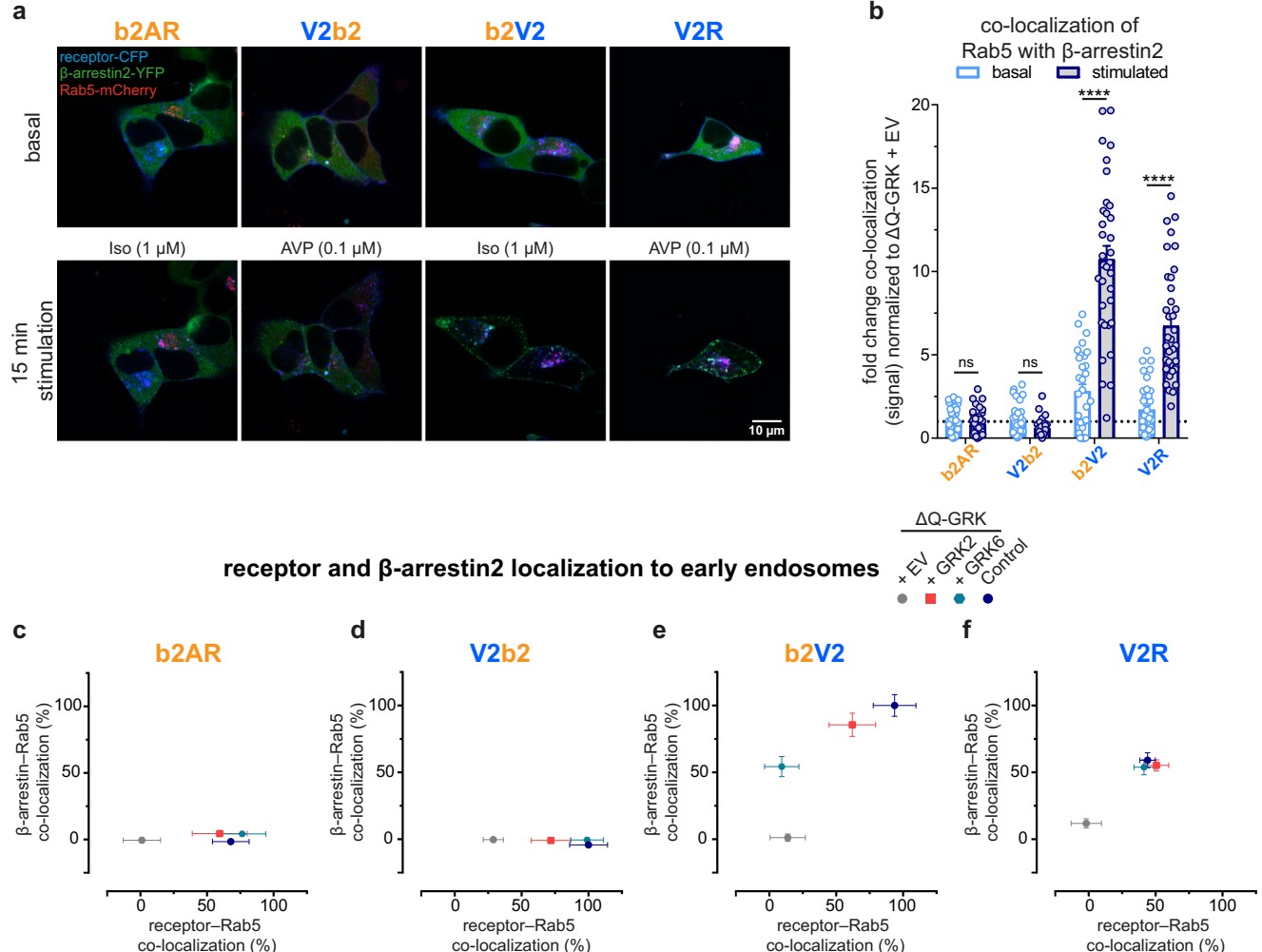

**Fig. 5 | β-arrestin translocation to early endosomes is controlled by the receptor C-terminus.** Control and ΔQ-GRK cells, overexpressing GRK2, GRK6 or no GRKs (empty vector (EV)-transfected) were transfected with the indicated receptor-CFP, β-arrestin-YFP and early endosome marker Rab5-mCherry. Confocal images were taken before (basal) and after 15 min of specified ligand.
**a** Representative images for all receptors, expressed in Control cells. **b** The co-localization of Rab5 with β-arrestin2 in Control cells was quantified in over 28 images for each condition using Squassh and SquasshAnalyst[44,45]. Data are presented as mean fold change in co-localization (signal) + SEM, normalized to the respective unstimulated (basal) ΔQ-GRK + EV condition. Statistical comparison between basal and stimulated values was performed using a two-way ANOVA, followed by a Sidak's test (ns not significant; ****$p < 0.0001$). Detailed results, also for

the two-way ANOVA, followed by a Tukey's test to compare basal and stimulated values between different conditions, can be accessed in Supplementary Tables 3–5. **c**–**f** Analogous to the data shown in (**b**), the fold change in co-localization (signal) was analyzed for the receptor or β-arrestin2 with Rab5, as fold change over the respective basal ΔQ-GRK + EV condition for each receptor and GRK condition (Control, ΔQ-GRK + EV, +GRK2, +GRK6). The stimulated values for receptor–Rab5 co-localization are show on the x-axis and for β-arrestin2–Rab5 co-localization on the y-axis. For both dimension, the data points were normalized to the respective maximum (for receptor–Rab5 co-localization: V2b2 in Control cells, for β-arrestin2–Rab5 co-localization: b2V2 in Control cells) and are shown in percent. Source data are provided as a Source Data file.

Supplementary Fig. 10). All of the receptor variants show a significant translocation to early endosomes upon stimulation in Control cells, co-transfected with β-arrestin2, while none of the tested GPCRs showed a significant increase in Rab5-co-localization in the absence of GRKs (Supplementary Fig. 9a, b, Supplementary Table 4). However, only receptors featuring the V2R C-terminus were able to mediate the delivery of β-arrestin2 to early endosomes in presence of GRKs (Fig. 5e, f, Supplementary Fig. 10b, c). In contrast, the b2AR and V2b2 receptors did not display a significant endosomal translocation of β-arrestin2 (Fig. 5c, d, β-arrestin1–V2b2 in Supplementary Fig. 10a). The GRK2- and 6-specific conditions were overall similar to the measurements in Control cells in all but one case, as GRK6 alone failed to induce b2V2 translocation to subcellular compartments that harbor Rab5 (Fig. 5e). Thus, our data show that in contrast to ERK1/2 phosphorylation, the early trafficking of GPCRs and β-arrestins is primarily orchestrated by the C-terminus of the receptor.

## Relating GPCR domains with functional outcomes via transferability coefficients

Through our comprehensive analysis, we are now able to address a central question in β-arrestin biology and delineate which aspects of receptor regulation are induced by the transmembrane helix-bundle, the C-terminus or on an individual combination of both. To this end, we introduce transferability coefficients (Supplementary Fig. 11), which distinctly quantify the influence of the receptor moieties on assessed β-arrestin characteristics. To determine these coefficients based on our β-arrestin2 conformational change data, we first normalized the datasets to the maximum signal within a receptor-specific conformational fingerprint (Supplementary Fig. 11a), following the comprehensive pharmacological analysis used to identify non-responding β-arrestin regions and conditions (Supplementary Fig. 2). Further, we examined differences between the wild type receptors (b2AR and V2R) and compared the results for each chimera (b2V2 and V2b2) to the

corresponding wild type receptor featuring either the same helix-bundle or C-terminus. For a pronounced difference between receptors sharing the same transmembrane helix-bundle, the calculation yields high *transferability$_{tail}$* values and vice versa for the *transferability$_{core}$*. Subsequently, *transferability$_{core}$* was subtracted from the *transferability$_{tail}$* value (Supplementary Fig. 11b). Using this strategy, a positive coefficient indicates that conformational changes in a particular β-arrestin region are mostly induced by the GPCR C-terminus, or more accurately, "C-terminus transferable", as exemplified by β-arrestin2-F4 in Control cells. In contrast, a negative value suggests "helix-bundle transferability", as seen for β-arrestin2-F10 (Supplementary Fig. 11c).

In our comparison of β-arrestin2 conformational changes upon interaction with the b2AR or V2R in Control cells, we found differences in rearrangements of the phosphorylation-sensing N-domain (Fig. 1g–j), indicating a more pronounced engagement of β-arrestin2 with the V2R C-terminus in comparison to binding of the b2AR counterpart. Similarly, examination of C-domain conformational changes suggested differences in membrane association of the β-arrestin2 C-edge loops[46] when coupling to b2AR or V2R. These data are consistent with previous observations that class A and B GPCRs elicit distinct β-arrestin conformational changes[12], induced by different complex configurations.

Regarding β-arrestin1, despite a similar recruitment (Supplementary Fig. 1g, h), the conformational change biosensors demonstrated a vastly different fingerprint for the interaction with the V2R, when compared to β-arrestin2 (Supplementary Fig. 1m, n). This was also evident for other receptors, such as the parathyroid hormone 1 receptor, which induced distinct conformational rearrangements in the two β-arrestin isoforms[25].

Interestingly, we were unable to detect β-arrestin1 conformational changes upon interaction with the b2AR (Supplementary Fig. 1i, j), despite observing its direct recruitment to the b2AR to some extent (Supplementary Fig. 1e, f). This is consistent with previous reports, which indicate that β-arrestin1 exceedingly relies on C-terminus interactions, whereas β-arrestin2 is better suited to utilize the helix-bundle-binding interface[25,26]. The interaction interface of the b2AR appears less conducive to β-arrestin1 binding, aligning with the original reports that class A receptors exhibit a lower affinity for β-arrestin1[17].

Comparing wild type and chimeric receptor-induced β-arrestin conformational changes using the transferability coefficient, we noticed a consistent trend of N-domain conformational changes (F2, F4, F5) being C-terminus transferable under endogenous GRK expression (Fig. 6a–d). We showed that these are predominantly GRK-dependent (Fig. 2s–v). This phenotype was particularly pronounced for position F4. Notably, in the complex crystal structure of the V2R with β-arrestin, the phosphorylated C-terminus closely interacts with sites adjacent to the F4 position. Given that the β-arrestin N-domain serves as the primary interaction site for phosphorylated receptor domains, it is not surprising that conformational changes in this region are strongly influenced by the specific GPCR C-terminus. Additionally, the β-arrestin N-domain harbors numerous known interaction sites with members of the MAPK family[47–49]. It is tempting to speculate that the observed differences in N-domain conformational changes, particularly in a receptor C-terminus-dependent manner, may impact the scaffolding of MAPKs, yet our data neither confirms nor refutes this notion. In contrast, responses of most FlAsH sites located in the β-arrestin C-domain were influenced by the unique combination of the receptor helix-bundle and C-terminus. Of note, we were only able to measure conformational changes in the C-edge region (F1) for β-arrestin2 binding to the V2R. Here, it is most likely that this receptor induces pronounced different interaction of β-arrestin2 with the plasma membrane[46], as a result of the specific configuration of the V2R. This difference aligns with a previous study showing that the

dependence of the GPCR–β-arrestin interaction on an additional membrane anchor corresponds to the class A/class B classification in terms of β-arrestin engagement[50]. However, C-domain-localized position F10 showed a pronounced helix-bundle transferability (Fig. 6a–d). Lee et al. were able to correlate conformational changes at this position (insertion at amino acid 263, termed F5 in their publication) with ERK1/2 activation[51]. As this signaling is initiated by G proteins[15], which mainly interact with the receptor transmembrane helix-bundle, our findings suggest that GPCR helix-bundles orchestrate both G protein-, as well as β-arrestin-facilitated differences in ERK1/2 activity.

## Discussion

In this study, we systematically investigated the influence of specific receptor domains, namely, the transmembrane helix-bundle, including its connecting loops, and the C-terminus, on various aspects of GPCR signaling and regulation: receptor phosphorylation (Figs. 2 and 4), interaction with β-arrestins (Figs. 1 and 2) and their conformational rearrangements (Figs. 1, 2 and 4), as well as ERK1/2 phosphorylation (Fig. 3) and receptor internalization (Fig. 5). To achieve this, we employ the b2AR and the V2R as model class A and B receptors, alongside their respective transmembrane helix-bundle and C-terminal chimeras, V2b2 and b2V2.

In addition to our overall evaluation of GPCR domain-induced β-arrestin conformations (Fig. 6a, b), we observed GRK-specific differences in β-arrestin conformational changes. Overexpression of GRK2 or 6 in knockout cells devoid of endogenous GRK2/3/5/6 expression, revealed distinct conformational rearrangements for β-arrestin–GPCR interactions in presence of either GRK2 or 6 (Fig. 4i–l, Supplementary Fig. 5). Furthermore, we identified different dependencies on the investigated receptor domains for GRK2- and GRK6-mediated conformational rearrangements in β-arrestins (Fig. 4m, n, Supplementary Fig. 12). Specifically, β-arrestin conformational fingerprints in presence of GRK2 clustered according to similarity in a receptor C-terminus-dependent manner, whereas no such trend was observed for β-arrestin conformational changes in presence of GRK6 alone. Of note, for receptors sharing the V2R transmembrane helix-bundle, the GRK6-dependent β-arrestin2 conformational changes were similar. This was not observed for β-arrestin1 or receptors sharing the b2AR transmembrane helix-bundle, which indicates receptor-specific differences in the utilization of GRK and β-arrestin isoforms.

We further explored the potential functional outcomes of differential GPCR–β-arrestin interactions and quantified the contribution of specific receptor domains using the described transferability coefficient (Supplementary Fig. 11). Employing GRK2/3/5/6 knockout cells, we expanded our analysis to investigate GRK-dependent and GRK-specific influences on the ERK1/2 activation profile, receptor phosphorylation, β-arrestin recruitment and internalization phenotype (Fig. 6e, f).

The temporal profile of ERK1/2 activation clearly exhibited helix-bundle transferability. Furthermore, we observed a helix-bundle-dependence of conformational rearrangements at position F10 in the β-arrestin2 C-domain, a site previously associated with ERK1/2 activation[51]. Here, the presence or absence of endogenous GRKs did not alter the kinetic profile of ERK1/2 signaling. Moreover, our analyses of the four different receptor variants did neither allow us to confirm nor refute GRK-dependent effects that are consistent with a supportive role of β-arrestins in the activation of ERK1/2 (Fig. 3, Supplementary Fig. 4)[52–56]. This contrasts with an older study, in which the authors observed a shift from a prolonged ERK activation profile, similar to what we found for the V2R, to a sharp peak at around two minutes when stimulating the angiotensin II type 1 receptor under β-arrestin2 knockdown conditions using siRNA, resembling the kinetic profile we observed for the b2AR[34]. The authors deduced that the sharp peak around two minutes would be G protein-dependent, while the prolonged phase would be β-arrestin2-dependent. By now, it has become

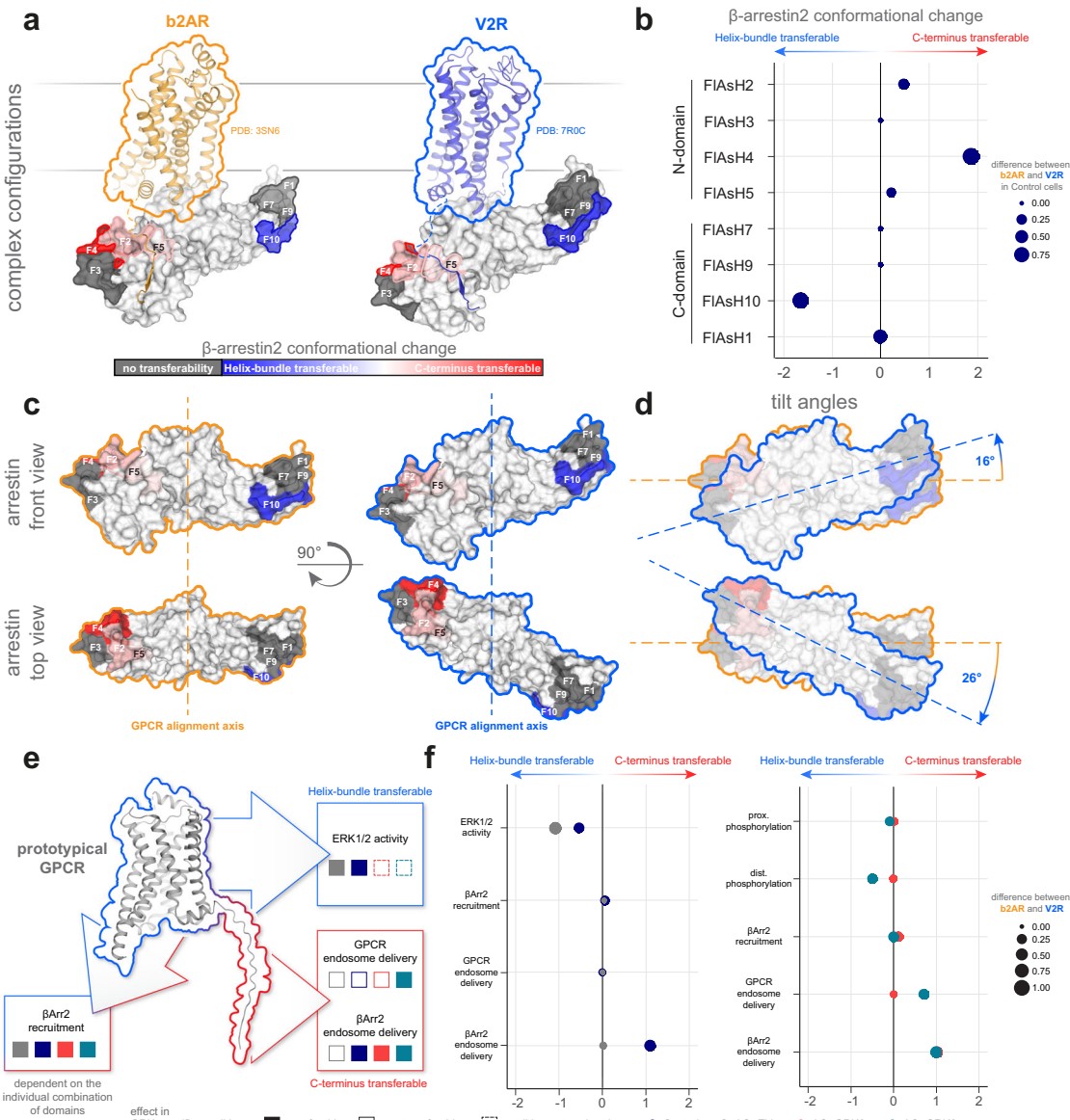

**Fig. 6 | Distinct aspects of receptor–β-arrestin2 interactions are differentially influenced by GPCR transmembrane helix-bundles or C-termini. a** Complex configurations for β-arrestins with an adrenergic receptor (β1 adrenergic receptor (b1AR), here exchanged by the aligned structure of active b2AR, PDB: 3SN6) and the V2R (PDB: 7R0C) are shown. Both complex structures feature β-arrestin2 aligned in place of β-arrestin1, while calculated transferability coefficients (see Supplementary Fig. 11 and methods) of β-arrestin2 biosensors in Control cells are projected onto its surface structure (PDB: 3P2D) to indicate the influence of individual GPCR domains on conformational changes (blue indicates helix-bundle transferability of conformational changes, while shades of red indicate C-terminus transferability and biosensors that do not show any transferability are colored in gray). **b** Calculated transferability coefficients of all β-arrestin2 biosensors in Control cells are shown as bubble plots, also indicating the initial difference in signal between WT GPCRs, shown as the size of the plotted symbols. **c, d** β-arrestin configurational positioning and tilt angles (with respect to the GPCR alignment axis and plasma membrane) are compared between complexes formed with an adrenergic receptor (orange outline) and the V2R (blue outline). **e, f** Calculated transferability coefficients are depicted for functional assay data, featuring ERK1/2 activity, β-arrestin2 recruitment, GPCR and β-arrestin2 endosomal delivery, as well as proximal and distal, C-terminal GPCR phosphorylation, in all GRK-specific conditions. These coefficients are shown schematically in (**e**), and as bubble plots in (**f**), while the size of the plotted symbols indicates the initial difference in signal between WT GPCRs. Source data are provided as a Source Data file.

evident that in the complete absence of G proteins, no β-arrestin-mediated ERK activation is measurable in living cells[15]. Since the overall kinetics of ERK activation did not change in our experiments, we cannot conclude that the GRK-mediated β-arrestin interaction is solely responsible for the prolonged phase.

As one key receptor modification, shown to orchestrate specific β-arrestin interactions and resulting functional outcomes[57–59], we investigated receptor phosphorylation at two specific clusters per receptor C-terminus (proximal and distal). To accomplish this, we created cell lines that stably overexpress individual receptors, as well as GRK2 or 6.

The utilized, YFP-tagged GRK constructs, facilitated cell sorting based on comparable fluorescence intensity to ensure similar expression levels. Relative to endogenous GRK expression, we generally observed higher levels of phosphorylation in presence of GRK6 as compared to GRK2 (Fig. 4a–h). Notably, while we sorted the stable GRK2- and 6-overexpressing cell lines for similar YFP signals, the divergent sub-cellular localization of these GRK isoforms possibly results in a higher local concentration of GRK6 in proximity to GPCRs at the plasma membrane. This, in combination with the different GRK recruitment mechanisms[37–41], may account partially for the relatively lower

efficiency of GRK2 in comparison to GRK6, as shown in our measurements. With respect to the observed transferability, we only found slight helix-bundle transferability in case of GRK6-dependent distal receptor phosphorylation (Fig. 6e, f). We attribute this to exceeding GRK6-mediated phosphorylation observed for receptors sharing the b2AR helix-bundle, similar to the receptor phosphorylation measured under endogenous expression in Control cells – an observation interestingly not mirrored by V2b2 and V2R (Fig. 4a–h, Supplementary Table 2). Of particular interest, the C-terminal GPCR phosphorylation pattern was not exclusively influenced by the C-terminus. While this observation might seem counterintuitive at first, it aligns with existing GRK–GPCR complex structures, which indicate that GRK interactions are facilitated by the insertion of the N-terminal GRK helix into the active GPCR helix-bundle[3,42]. However, the hierarchy of events in GRK–GPCR interactions remains underexplored and could differ for membrane-associated (GRK5 and 6) and cytosolic GRKs (GRK2 and 3).

In the context of measured β-arrestin2 recruitment and GPCR endosomal delivery, no definitive transferability between receptor domains was identifiable, except for GRK6-mediated receptor internalization, which displayed a tendency towards C-terminal transferability. This suggests that these processes are governed by the specific combination of receptor domains.

Conversely, endosomal delivery of β-arrestin2 clearly showed to be dependent on GRKs and exhibited a pronounced C-terminus transferability, a pattern consistent across all GRK-expressing conditions (Con, GRK2, GRK6) (Figs. 5 and 6e, f). This finding corresponds well with initial reports highlighting the pivotal role of the C-terminus in the observed GPCR–β-arrestin co-internalization phenotype of class A or B receptors[16]. Based on our data, we can now confirm that endosomal β-arrestin complex formation is controlled via GRK-dependent phosphorylation sited within GPCR C-termini.

By focusing on two of the most prominent class A and B receptors, b2AR and V2R, as well as their chimeras, we have discerned that N-terminal β-arrestin2 conformational change patterns are primarily C-terminus transferable under endogenous GRK expression in HEK293 cells. In contrast, molecular rearrangements in the C-domain were dependent on the transmembrane helix-bundle, namely at position F10, previously correlated with ERK activation[51]. In line, the temporal profile of ERK1/2 phosphorylation also demonstrated helix-bundle transferability, while β-arrestin delivery to early endosomes was clearly dependent on the receptor C-terminus. Taken together, these findings significantly advance our understanding of the GPCR structure-function relationship, offering valuable insights into the complex interplay between receptor domains and downstream signaling pathways.

## Methods

### Cell culture
CRISPR/Cas9-generated GRK2/3/5/6 knockout (ΔQ-GRK) and the respective Control cells with unaltered GRK expression[26] were cultured in Dulbecco's Modified Eagle's Medium (DMEM; Sigma-Aldrich #D6429), containing 10% fetal calf serum (FCS; Sigma-Aldrich #F7524) and 1% penicillin and streptomycin mixture (Sigma-Aldrich #P0781) at 37 °C with 5% $CO_2$. Cells were passaged every 3–4 days and regularly checked to be negative for mycoplasma infection using the Lonza MycoAlert mycoplasma detection kit (LT07-318).

### Generation of stable cell lines
Control and ΔQ-GRK cells were transfected with pcDNA3 plasmids containing hemagglutinin (HA)-tagged b2AR, V2b2, b2V2 or V2R utilizing polyethylenimine (PEI) transfection reagent (Sigma-Aldrich, 408727, diluted to 10 μg/ml, pH 7.2 adjusted with HCl). Transfected cells were selected by adding 0.5 g/l G418 (Capricorn Scientific, #G418-Q). After selection was completed, the cells were continuously cultivated in cell culture growth media containing 0.2 g/l G418. The cells

were sorted for equal receptor surface expression in cells expressing receptors sharing the same transmembrane helix-bundle via fluorescence-activated cell sorting (FACS) utilizing an AF647 coupled anti HA-tag antibody (Biolegend, #682404). Equal expression was validated via flow cytometry and Western blot analysis.

To create the cell lines, stably expressing human GRK2-YFP or human GRK6-YFP in addition to each receptor, the retroviral expression vectors pMSCV-huGRK2-YFP-IRES-Puromycin and pMSCV-huGRK6-YFP-IRES-Puromycin were created. Freshly produced retroviral particles were used to transduce the before mentioned ΔQ-GRK cells, stably expressing HA-b2AR, HA-V2b2, HA-b2V2 or HA-V2R. The cells were selected with 0.8 μg/ml Puromycin (Sigma-Aldrich, P8833) and sorted for similar GRK expression levels via the coupled YFP. All cell lines are available from the authors upon request.

### Western blots
For analysis of ERK phosphorylation, $4.5 \times 10^6$ Control cells or $6 \times 10^6$ ΔQ-GRK cells stably expressing HA-tagged b2AR, V2b2, b2V2 or V2R were seeded into 58 cm² dishes 24 h before transfection. To avoid confusion between b2AR and β-arrestin2, we refrain from referring to the receptor as $β_2AR$ and consistently write it as b2AR and b2V2 for the chimera. The cells were transfected with 10 μg empty vector (EV), β-arrestin1 or β-arrestin2 using PEI transfection reagent (Sigma-Aldrich, 408727, diluted to 10 μg/ml, pH 7.2 adjusted with HCl). After overnight incubation, $1.8 \times 10^6$ Control cells and $2.8 \times 10^6$ ΔQ-GRK cells were seeded into poly-D-lysin-coated 6-well plates. After 24 h, the cells were starved from FCS for 4 h and subsequently stimulated with either 1 μM Iso (b2AR, b2V2) or 100 nM AVP (V2R, V2b2) for indicated time points or left untreated. Then, cells were washed with ice-cold PBS and lysed with RIPA lysis buffer (1% NP-40, 1 mM EDTA, 50 mM Tris pH 7.4, 150 mM NaCl, 0.25% sodium deoxycholate), supplemented with protease and phosphatase inhibitor cocktails (Roche, #04693132001, #04906845001). Cleared lysates were supplemented with sample buffer and heated for 30 minutes at 50 °C. Equal amounts of protein were subjected to polyacrylamide gel electrophoresis and analyzed for vinculin (BIOZOL, BZL03106; 1:1000), pERK (phospho-p44/42, Cell signaling technology, #9106; 1:1000) or total ERK (p44/42, Cell signaling technology, #9107; 1:1000) as indicated. Goat anti-rabbit (SeraCare, #5220-0336; 1:10,000) and goat anti-mouse (SeraCare, #5220-0341; 1:10,000) antibodies were utilized for primary antibody detection. Quantification was performed using Fujifilm Multi Gauge V3.0 software. The pERK value for each condition was divided by its corresponding total ERK value.

### 7TM phosphorylation assay
The following antibodies were obtained from 7TM Antibodies (Jena, Germany): the phosphosite-specific antibodies against b2AR anti-pS355/pS356-β2 (7TM0029A) and anti-pT360/pS364-β2 (7TM0029B), the phosphosite-specific antibodies against V2R anti-pT359/pT360-V2 (7TM0368B) and anti-pS362/pS363/pS364-V2 (7TM0368C) as well as the rabbit polyclonal anti-HA antibody (7TM000HA).

Phosphorylation of b2AR, V2b2, b2V2 and V2R was assessed using the 7TM phosphorylation assay as previously described by Kaufmann et al.[30]. Control or ΔQ-GRK cells, stably expressing one of the receptors and GRK2 or 6, as indicated, were seeded into poly-L-lysine-coated 96-F-bottom-well cell culture microplates (Greiner Bio-One, 655180) and grown to >95% confluency. After the cells were exposed with increasing concentrations of either Iso or AVP (Tocris, 1747 and 2935) for 30 min at 37 °C, they were lysed with a detergent buffer (150 mM NaCl; 50 mM Tris-HCl, pH 7.4; 5 mM EDTA; 1% Igepal CA-360; 0.5% deoxycholic acid; 0.1% SDS) containing protease and phosphatase inhibitors (Roche, cOmplete mini #04693132001 and PhosSTOP #04906845001). When transferring the lysate into 96-U-bottom-well assay plates (Greiner Bio-One, 650101), it was split into two corresponding wells which subsequently allowed the detection of

phosphorylated and total receptor levels of each sample. Mouse anti-HA magnetic beads (Thermo Fisher, 88837) were used to enrich HA-tagged proteins. Phosphosite-specific antibodies (pS355/pS356-β2, pT360/pS364-β2, pT359/pT360-V2, pS362/pS363/pS364-V2) or phosphorylation-independent anti-HA (7TM Antibodies) were added as primary antibodies and incubated overnight. Anti-rabbit HRP-linked antibody (Cell Signaling Technology #7074) served as secondary antibody. Therefore, a color reaction could be initiated with the addition of Super AquaBlue detection solution (Thermo Fisher, 00-4203-58), which was stopped with 0.625 M oxalic acid. Magnetic beads were pulled-down by applying magnetic force to the plate and the supernatant was transferred to a transparent 96-well F-bottom detection plate (Greiner Bio-One, 655182). The optical density at 405 nm was measured by the FlexStation 3 microplate reader (Molecular Devices) and data were acquired through the SoftMax Pro 5.4 software. Further calculations and normalizations were performed in Excel 16.0. Therein, the background signal was subtracted from the raw OD values and every phosphorylation signal was corrected according to its corresponding loading control. Finally, the data was normalized to the negative (unstimulated) and positive control (maximal agonist concentration) on every plate. Concentration-response curves were generated using GraphPad Prism 9.3.1 software. All graphs display mean ± SEM of $n = 5$ independent experiments performed in duplicates.

### Intermolecular bioluminescence resonance energy transfer (BRET)

The β-arrestin recruitment to the V2b2 was performed and analyzed as for the b2AR, b2V2 and V2R in Drube et al.[26]. Briefly, cells were transfected with 1.5 μg V2b2, C-terminally fused to a Halo-ligand binding Halo-tag and 0.375 μg of β-arrestin, C-terminally coupled to a Nano-Luciferase (NLuc) in 21 cm² dishes, following the Effectene transfection reagent manual (Qiagen, #301427). The following day, 40,000 cells per well were seeded into white poly-D-lysine-coated 96-well plates (Brand, 781965) in presence of Halo-ligand (Promega, G980A) at a ratio of 1:2000. For each transfection, technical replicates were seeded as triplicates and a mock labeling condition without Halo-ligand was included. Before the measurement the next day, cells were washed twice with measuring buffer (140 mM NaCl, 10 mM HEPES, 5.4 mM KCl, 2 mM CaCl₂, 1 mM MgCl₂; pH 7.3) and NLuc-substrate furimazine (Promega, N157B) was added (1:3500 in measuring buffer). The measurements were performed utilizing a Synergy Neo2 plate reader (Biotek), operated with the Gen5 software (version 2.09), with a custom-made filter (excitation bandwidth 541–550 nm, emission 560–595 nm, fluorescence filter 620/15 nm). After 3 min baseline monitoring, the indicated concentration of AVP was added and the measurements were continued for 5 min. We corrected the initial BRET ratio for labeling efficiency by subtracting the values measured for mock labeling conditions. The technical replicates were averaged for each measurement. By the division of the corrected and averaged values, measured after ligand stimulation by the respective, corrected and averaged baseline values, we calculated baseline-normalized, labeling-corrected BRET changes. Subsequently, this corrected BRET change was divided by the vehicle control for the final dynamic Δ net BRET change. For the concentration-response curves, the data points 2–4 min after stimulation were averaged. All concentration-dependent curves were normalized to the lowest concentration to display the maximal measured range. These calculations were performed using Excel 16.0.

### Intramolecular BRET

Control or ΔQ-GRK cells were transfected in 21 cm² dishes with 1.2 μg of untagged receptor, 0.12 μg of the respective β-arrestin biosensor and 0.25 μg of GRK2 or GRK6, as indicated, adjusted to the total DNA amount of 2 μg with EV, according to the Effectene transfection reagent

protocol (Quiagen, #301427). The following day, 40,000 cells per well were seeded into poly-D-lysine-coated white 96-well plates (Brand, 781965). The FlAsH labeling and β-arrestin conformational change measurements on the next day were performed as described in Haider et al.[25]. In short, cell culture growth media was aspirated and cells were washed twice with PBS, then incubated with 250 nM FlAsH in labeling buffer (150 mM NaCl, 10 mM HEPES, 25 mM KCl, 4 mM CaCl2, 2 mM MgCl2, 10 mM glucose; pH7.3) with 12.5 μM 1,2-ethane dithiol (EDT) for 60 min at 37 °C. After aspiration of the FlAsH labeling or mock labeling solutions, the cells were incubated for 10 min at 37 °C with 100 μl per well of 250 μM EDT in labeling buffer. Addition of NLuc substrate, measurement and analysis were conducted as described above.

### Pharmacological analysis of β-arrestin conformational changes and bioinformatical clustering

The normalized net BRET changes were subjected to thorough pharmacological analysis (Supplementary Fig. 2). Briefly, concentration-dependent net BRET changes were fitted using a non-linear four-parameter model, while fitting parameters (EC₅₀ and Hill slope) were extracted and plotted to define appropriate constrains for concentration-dependent response of conformational change sensors. To achieve a sigmoidal fit with the concentrations of ligand applied, it is necessary for the absolute value of the Hill slope to exceed 0.1, and for the EC₅₀ to fall within the range of $10^{-3}$ to $10^{0.3}$ μM. According to these, the datasets were classified as either non-responding or responding. Similarly, experiments for which no fit could be defined were considered as non-responding. Additionally, four individual datasets were manually defined as non-responding, due to the lack of concentration-dependent signal change. Non-responding sensors were assigned the value of zero net BRET change.

Data were preprocessed using Python 3.11, while subsequent analysis and classification was conducted in R 4.2.2. The four-parameter fit was generated using *drc* R package (R package version 3.0-1)[60]. Clustering heatmaps in Fig. [4]m, n were generated using the pheatmap R package (Kolde, R. (2013). pheatmap: Pretty Heatmaps. R package version 1.0.12. http://CRAN.R-project.org/package=pheatmap.).

### Confocal microscopy

Live cell microscopy was performed as described in Haider et al.[25]. Briefly, Control or ΔQ-GRK cells were transfected in 21 cm² dishes with 1 μg of receptor-CFP, 0.5 μg of β-arrestin1/2-YFP, 0.5 μg of Rab5-mCherry and 0.25 μg of GRK2 or GRK6, as indicated, adjusted with EV to the total DNA amount of 2.5 μg, according to the Effectene transfection reagent protocol (Quiagen, #301427). The next day, $1 \times 10^6$ cells per well were seeded in 6-well plates containing poly-D-lysine-coated glass cover slips. Before imaging, cells transfected with receptors containing the b2AR helix-bundle were starved for ~4 h. Coverslips were washed twice with measuring buffer and images were acquired before and after 15 min of indicated agonist stimulation utilizing the inverted laser scanning confocal microscope (DMi8 TCS SP8, Leica Application Suite X software, Leica microsystems) in a 1024 × 1024 pixel format, using a 63x water immersion objective, zoom factor 3, line average 3 and 400 Hz. The fluorophores were excited at 442 nm (CFP), 561 nm (mCherry) and 514 nm (YFP). The stimulation movies were recorded analogously over 15 min with an image taken every 30 s and cells were stimulated after the fourth frame. Co-localization of the fluorophore-coupled proteins was analyzed with the ImageJ-based software Squassh (segmentation and quantification of subcellular shapes) and R-based software SquasshAnalyst[44,45].

### Statistical analysis

Differences in the AUC of ERK1/2 phosphorylation, normalized OD 405 nm (%) of assessed receptor phosphorylation and fold changes in co-localization of the proteins quantified from the confocal images

were compared using Student's *t* test or two-way analysis of variance (ANOVA), followed by the adequate multiple comparisons, as described in the respective figure legends. Statistical analysis was performed using GraphPad Prism7.03, with a type I error probability of 0.05 considered to be significant.

## Quantification of helix-bundle and C-terminus transferability

All results generated with wild type and chimeric receptors were compared to quantify helix-bundle and C-terminus transferability (Supplementary Fig. 11). Briefly, C-terminus transferability (transferability$_{tail}$) was quantified as the sum of the absolute differences between responses of the wild type GPCRs and the respective chimeric receptor featuring the same helix-bundle but a different C-terminus. In turn, helix-bundle transferability (transferability$_{core}$) was quantified as the sum of the absolute differences between responses of the wild type GPCRs and the respective chimeric receptor featuring the same C-terminus but different helix-bundle. The transferability coefficient was subsequently calculated as the unitless difference between tail- and core-transferability. Thus, a negative transferability coefficient suggests helix-bundle transferability of measured responses, while a positive coefficient indicates that individual receptor signaling behaviors are C-terminus transferable. Additionally, the absolute difference between wild type GPCRs was calculated and used to scale data points in the bubble plot presentation of transferability coefficients. Calculations and plotting were conducted in R 4.2.2.

To achieve comparability between the different GPCR signaling assays, datasets were normalized prior to calculating the transferability coefficient. Conformational change datasets were additionally normalized to the maximum signal for each receptor. Proximal and distal C-terminus phosphorylation measurements were normalized as described above. The β-arrestin recruitment and confocal microscopy-derived datasets were normalized to the basal signal in the respective ΔQ-GRK condition and the maximum signal for each receptor. ERK1/2 phosphorylation was quantified as area under the curve (AUC) and normalized to the signal of the time point exhibiting the maximal response in Control cells for each GPCR, respectively.

### Reporting summary

Further information on research design is available in the Nature Portfolio Reporting Summary linked to this article.

## Data availability

The data that support this study are present within the manuscript and supplementary information files. Source data are provided in this paper. Source data are provided with this paper.

## Code availability

Custom code was created to enable the analysis of conformational change data, calculation of the transferability coefficient as well as the plots presented in Fig. 4m, n, Fig. 6b, f and Supplementary Fig. 2 and 11. The large language model Chat GPT-4 was utilized to optimize code (OpenAI. (2024), ChatGPT-4 [Large language model]. https://chat.openai.com). The code is available via Github (https://github.com/moyoda/Matthees_fingerprints).

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

## Acknowledgements

C.H. was supported by the European Regional Development Fund (Grant ID: EFRE HSB 2018 0019). R.S.H. was partially funded by the Deutsche Forschungsgemeinschaft (DFG, German Research Foundation; grant ID: 519415594). C.H. and L.K. were additionally supported by the federal state of Thuringia and the Deutsche Forschungsgemeinschaft (Grant: *Polytarget*; SFB1278: 316213987, project D02). J.D. was additionally funded by the University Hospital Jena IZKF (Grant ID: MSP10). We thank Dr. J. Robert Lane for constructive discussions and pharmacological evaluation of the results presented in this manuscript, Prof. Tom Kirchhausen for providing the Rab5-mCherry plasmid and the Core Facility Flow cytometry of the FLI–Leibniz Institute for Age Research, Jena, for sorting of the stable cell lines.

## Author contributions

E.S.F.M., R.S.H. and C.H. developed the concept and designed experiments; E.S.F.M., R.S.H., L.K., M.R., V.W., T.T., C.Z. conducted experimental work; N.K.B. performed 7TM phosphorylation analysis, supervised by S.S.; J.D., L.K., V.W., E.S.F.M. created stable cell lines; E.S.F.M., R.S.H, M.R. and L.K. compiled the data; M.R. performed bioinformatics analysis of non-responders/responders conformational change conditions and

transferability coefficients; C.H. supervised the project; E.S.F.M., R.S.H., L.K., M.R. and C.H. wrote the paper; all other authors critically revised the paper and gave final approval.

## Funding

## Competing interests
S.S. is the founder and scientific advisor of 7TM Antibodies GmbH, Jena, Germany. The remaining authors declare no competing interests.
