## [Transparent Peer Review file · Nature Communications]

Helix-bundle and C-terminal GPCR domains differentially influence GRK-specific functions and β -arrestin-mediated regulation

Corresponding Author: Professor Carsten Hoffmann

Version 0:

Reviewer comments:

Reviewer #1

(Remarks to the Author)

In this report, the authors try to correlate beta arrestin conformational changes with 2 GPCRs with distinct beta arrestin recruitment modalities, as well as their C-terminal exchange chimeras, with the capacity to elicit ERK1/2 signalling and the GRK requirement by using a cell line with deleted GRKs, as well as other 2 lines re-expressing GRK2 and GRK6.

I found the report quite interesting although the writing style could be improved to enhance its readability. The introduction could be written in a more engaging style for example. Some data shown in figures has already been published elsewhere and should not be included there but rather just cited. In terms of the Figures, I found Supplementary Figure 2 quite confusing. The conformational sensor they show as example keeps changing in Supplementary Figure 2a - I believe they are trying to explain their classification of the conformational effects but I still found this highly confusing.

For Figure 2, I can see there are changes in the BRET data for the F1 sensor, but these are not further commented in the results - the authors should discuss these changes.

The ERK1/2 data raises some technical issues: The authors analyzed ERK1/2 phosphorylation in Control and Δ Q-GRK cells stably overexpressing one of the receptor constructs. However, there could be clonal differences in responses simply due to not using the same background cells to analyse this. At the very least, the authors need to perform rescue experiments where they re-express the GRKs and see if the ERK1/2 responses now switch back to those of cells not deleted for GRK. This experiment is key to be able to exploit the results from this analysis - otherwise the results are not reliable.

The ERK1/2 results with Barr overexpression are overinterpreted: Overexpression of barr1 has no significant effect on ERK phosphorylation and that of barr2 only has an effect in chimeric receptors, not in normal ones. The authors need to state this and tone down their assumptions in terms of the role of barr1/2 in ERK1/2 signalling, or perform alternative experiments (with barr1/2 KDs) that support their claims.

For the endocytic localisation data, the kinetics are not taken into account as only a time point has been imaged. Time lapse microscopy experiments would be more suitable to look at this. Additionally, in Fig 5 - how does barr2 reach the endosome without b2V2 for GRK6 expressing cells? Please give an explanation for these puzzling results or at least an hypothesis of what you think it is going on.

(Remarks on code availability)

Reviewer #2

(Remarks to the Author)
Review for Nature Communications

Edda Matthees et al. "Relating GPCR domains with functionality: receptor helix-bundle and C-terminus differentially influence GRK-specific functions and beta-arrestin-mediated regulation"

In their interesting manuscript, the Hoffmann team presents a plethora of data on various downstream effects of two prototypical GPCRs, the beta2-adrenergic (beta2AR) and the V2 vasopressin (V2R) receptors, plus two constructs where the C-termini have been swapped. A whole set of downstream effects is measured: GRK-dependent phosphorylation of proximal and distal C-terminal domains (including GRK subtype-selectivity), beta-arrestin recruitment (including beta-arrestin1 vs. -2 selectivity), ERK1/2 activation, and receptor internalization (including subcellular localization and co-localization with rab5).

This results in a large array of data and possible correlations, and I have to confess that I had to read the manuscript several times to understand the main conclusions (and apologies that I needed the Christmas break to get this done!).

The key points that the authors aim to address is, whether specific downstream effects are mediated more via the receptor core (=helix-bundle) or its C-terminus. To be able to address this, they rely on two receptors considered as prototypes for tight (=class B, V2R) or transient (class A, 2AR) -arrestin binding. They further make use of the well-established observation that beta-arrestin1 generally forms less tight interactions with receptors than beta-arrestin2.

The manuscript contains a plethora of carefully obtained and well-documented data, and quite obviously there are limits to what one could wish. For example, having only one data point per log unit may seem little for curve fitting purposes – but if we consider the effort of obtaining such complex data and given the many downstream pathways that are investigated, I believe that the authors have made every reasonable effort.

However, before going into any details, I believe that the manuscript would significantly benefit from three changes:

- The authors seem to shy away a little from precise conclusions and remain vague both in the abstract and in the discussion (particularly towards the end of the discussion). Focusing on the most significant findings (rather than describing everything one sees in the data – particularly in the results section, but also in the discussion) and drawing more explicit conclusions would make this manuscript more concise and, in my view, more interesting.
- The use of just two receptors plus two C-tail swapped variants obviously limits the generalizability of the findings and conclusions. Contrasting this with my wishes expressed above, it is certainly difficult, but in my view possible, to be somewhat more careful in postulating general principles and yet, at the same time, draw more explicit conclusions from the experiments.
- The authors mention the publications on GPCR/beta-arrestin complex structures and they provide structural images in many of their figures. However, I see little link between these structures and the specific mechanisms proposed in this manuscript. I would suggest strengthening such links – if at all possible

Specific points:

1. In the abstract, I would suggest to be much more explicit about key differences observed, and to eliminate statements such as "some processes – while others...". This would allow the reader to start reading with the authors' main conclusions in mind.
2. In the kinetic experiments (Fig. 1) it is not clear at what time point agonist was added (at t=0?) and why the exponential part of the calculated curves begins at very defined time points. In some traces, there are no data near these time points, which makes curve fitting a little arbitrary.
3. In Fig. 2 I would suggest concentrating essentially on the radar plots, which summarize the large number of curves depicted in panels c-r. These radar plots seem to show that -arrestin conformational changes are mostly dictated by the receptor's C-terminus (V2R better than 2AR). Exceptions appear to be the agonist-dependent changes that are also seen with 2AR (notably F2, F3, F10).
4. The idea that the pattern of ERK activation (short vs. long lived) is mediated via the receptor core and is, hence G-protein-mediated, is an attractive hypothesis (Fig. 3). However, it might be appropriate to accommodate also the thoughts of those researchers who disagree with that notion. It might also be good to contrast ref 15 with those publications that argue for a major (and perhaps G-protein-independent) role of -arrestins in this process.
5. It seems appropriate, given the demanding experiments, to use only one example each of GRK2,3 and GRK5,6 in experiments on GRK specificity (Fig. 4) – but as in the case for the only two receptors (see above) it might be appropriate to add a word of caution that differences to the subtypes not tested might be possible. In Fig. 4, I would again concentrate on describing the radar plots. A specific interesting effect might be the observation that F4 responds very differently to the two receptor tails.
6. The fact that endosomal localization is so strikingly specific for the receptor C-tail (Fig. 5) is very interesting. Can the authors suggest a mechanistic explanation? And how does this fit with much less pronounced specificity for -arrestin conformational changes.
7. I like the use of the transferability concept in Fig.6 to assign specific functions to core vs. C-tail. As suggested above – can this be interpreted structurally?

Minor point:

The authors use the terms arrestin and beta-arrestin interchangeably and most of the times it is not clear to me why they use which expression. I know there is some confusion (and controversy) about the right terminology - but it would make sense to stay within one terminology. In this case, since all data are on beta-arrestins, I would suggest to call them beta-arrestins.

(Remarks on code availability)

Reviewer #3

(Remarks to the Author)

This paper examines the distinct roles that the C-tail and the central GPCR helical bundles play in driving beta-arrestin conformational changes following activation by canonical class A GPCR, i.e. the β 2-adrenergic receptor (β 2AR with subscripted 2) and class B GPCRs, i.e. the vasopressin V2 receptor. They use established approaches to assess β -arrestin recruitment and β -arrestin conformation and provide an interesting analytical framework in which to assess the roles of different phosphorylations sites in the receptor C-tails and distinct conformational affects on the N- and C-terminal domains of β -arrestin.

Comments

In general the experiments are well designed, well conducted and well analyzed. The use of the different receptor chimeras and the CRISPR KO lines is a strength. Some things the authors could address include:

Major:

- 1) I'm not sure the magnitude of BRET changes (as shown in Figure 1e,f) are meaningful as such. Are the levels of biosensor expression similar? I assume since these are intramolecular BRET constructs there is no issue of stoichiometry but it would have been a useful control to include in supplementary data. Different receptor expression could also dramatically contribute to this change of magnitude.
- 2) Can the authors explain in better detail the odd results in Figure 2l, n? Could it be more nuanced than a simple effect of different helical cores? Moreover, L199: "we were also able to monitor a residual GRK-independent β -arrestin2 recruitment to all receptor variants, except the V2b2..." How can the authors reconcile the absence of β arr2 recruitment by V2b2 in Δ Q-GRK condition (Figure 2i) with the change of β arr2 conformational change fingerprint reported in Figure 2t ?
- 3) Figure 3: For ERK activation, the authors normalized the data as fold change compared to the untreated cells (n=1), then established some comparisons between ctrl vs KO-GRK. However, there is no certainty the basal phosphorylation of ERK is equivalent for every condition (difference in constitutive GPCR activity for instance) Hence, does KO GRK really decrease ERK activation, or could it augment receptor basal activity (especially considering that overexpressing GRK increased basal phosphorylation in Fig 4)? Does arr overexpression really increase ERK activation, or in fact reduce receptor constitutive activity? Immunoblotting is a semi-quantitative method, as such, without comparing the basal levels between conditions on a same gel, any mechanistic claims the authors make about arr involvement in ERK activation appears over-interpretative (such as I51 "Moreover, we demonstrate that some β -arrestin-supported", I270 " β -arrestin-supported mechanism"...). Hence, the limitation in I554 "possibly also influencing initial GRK and arrestin interaction" should be expanded in the ERK section, and comments about arr involvement with ERK ... maybe be rephrased with consideration.
- 4) What does the grey scale indicate in the heatmap?
- 5) In Figure 4, could the additional feature beyond receptor phosphorylation per se be based in the presence or interaction with distinct pools of Gbetagamma subunits for GRK2 or GRK6?
- 6) Do distinct ligands for each receptor drive distinct effects? This might be an interesting addition to the story.
- 7) Comparing transferability coefficients between Figure 6B and Supp Figure 11, we see the pattern can vary drastically depending on GRK expression. This means there is no universal transferability pattern regarding β -arrestin2 conformation... As such, authors should correct their conclusion I558: "By focusing on two of the most prominent class A and B receptors, β 2AR and V2R, as well as their chimeras, we have discerned that N-terminal β -arrestin2 conformational change patterns are primarily C-terminus transferable under endogenous GRK expression IN HEK 293 cells".

Minor:

- 1) Although β -arrestin is correctly written, the authors refer to b2AR rather than 2AR.
- 2) Did they make 8 new lines for Figure 4 or sixteen? Is this because some had -arrestin1 or 2 as well?
- 3) L117 I117: "Using a refined set of β -arrestin conformational biosensors, we recently uncovered arrestin conformational fingerprints that are induced by ligand-activation of the presumably unphosphorylated parathyroid hormone 1 receptor (PTH1R), in the absence of GRKs25. This finding confirms that both binding interfaces, namely the GPCR transmembrane helix-bundle and C-terminus, influence the resulting active arrestin conformation" : The first sentence is an experimental approach, not a finding. Please insert the main result of (25) that is supporting the second sentence.
- 4) I506: Regarding ERK: "Since the overall kinetics of ERK activation did not change in our experiments, we cannot

conclude that the GRK-mediated β -arrestin interaction is solely responsible for the prolonged phase. However, other GRK-specific functions may also contribute to this phenomenon, such as internalization.”, GRK-specific appears paradoxical since there is no kinetic difference between control and KO-GRK.

(Remarks on code availability)
Not applicable.

Reviewer #4

(Remarks to the Author)
I co-reviewed this manuscript with one of the reviewers who provided the listed reports. This is part of the Nature Communications initiative to facilitate training in peer review and to provide appropriate recognition for Early Career Researchers who co-review manuscripts.

(Remarks on code availability)

Version 1:

Reviewer comments:

Reviewer #1

(Remarks to the Author)
The authors have responded well to my criticisms including with additional experiments and I am happy to support the publication of the manuscript

(Remarks on code availability)

Reviewer #2

(Remarks to the Author)
The authors have responded thoughtfully and in much detail to the reviewers' questions and suggestions and have altered their interesting manuscript accordingly.

The authors have also taken up the suggestion to be more explicit in their conclusions and interpretation (particularly in the abstract), but they also resist the temptation to become simplistic. While I would have wished to see even more explicit conclusions, I do understand that the biology may indeed be too complex for simple explanations.

(Remarks on code availability)

Reviewer #3

(Remarks to the Author)
Thanks for the thoughtful responses to my queries and comments. I am satisfied with all but one response and it was (and remains minor). Although beta-arrestin is correctly written with Greek symbol, the authors refer to b2AR rather than beta2AR- this is not right despite what the editor says.

(Remarks on code availability)

Reviewer #4

(Remarks to the Author)
I co-reviewed this manuscript with one of the reviewers who provided the listed reports. This is part of the Nature Communications initiative to facilitate training in peer review and to provide appropriate recognition for Early Career Researchers who co-review manuscripts.

(Remarks on code availability)

Reviewer #1 (Remarks to the Author):

In this report, the authors try to correlate beta arrestin conformational changes with 2 GPCRs with distinct beta arrestin recruitment modalities, as well as their C-terminal exchange chimeras, with the capacity to elicit ERK1/2 signalling and the GRK requirement by using a cell line with deleted GRKs, as well as other 2 lines re-expressing GRK2 and GRK6.

I found the report quite interesting although the writing style could be improved to enhance its readability. The introduction could be written in a more engaging style for example.

We thank the reviewer for the overall positive assessment of our work. In response to this comment, we reworked large parts of the introduction, with the aim to structure it in a more readable and engaging way. Hence, superfluous adjectives have been removed and cumbersome phrases have been changed, for example in lines 67ff. All changes are marked in yellow to facilitate a direct comparison.

Some data shown in figures has already been published elsewhere and should not be included there but rather just cited.

While we agree that previously published results should not appear front and center, we saw ourselves forced to include few key data points of Drube et al. 2022 (<https://doi.org/10.1038/s41467-022-28152-8>), specifically for experiments describing the kinase-specific recruitment of the receptor variants. This is the only case for Fig. 2 k, m and n, as well as Suppl. Fig. 1 a and c. Here, we clearly indicate when data has been already published to avoid confusion, but in our opinion, this is the only way to facilitate readability. It is vital to establish direct comparisons of these data sets, specifically as systematic comparisons represent the basis of our calculation of transferability.

In terms of the Figures, I found Supplementary Figure 2 quite confusing. The conformational sensor they show as example keeps changing in Supplementary Figure 2a - I believe they are trying to explain their classification of the conformational effects but I still found this highly confusing.

We thank the reviewer for raising these concerns regarding the didactic flow of Supplementary Figure 2. In order to improve readability, we adjusted the chart as follows: the first box now features no color and only one arrow, as it shows representative examples of data input. Headings above the first two panels now feature "input data" (first panel) and "negative examples", as well as "positive examples" (second panel) on the respective sides, to clearly illustrate the reason to include this figure – showing the quality of obtained data and our analysis pipeline.

Furthermore, we added the following sentence in the figure legend of Supplementary Figure 2: "featuring different example datasets for results that

were eliminated in different steps of the analysis pipeline” to clarify that we do not intend to show the workflow for one example but rather show the range of the data.

For Figure 2, I can see there are changes in the BRET data for the F1 sensor, but these are not further commented in the results - the authors should discuss these changes.

F1 is indeed an interesting position localized in β -arrestin C-edge, potentially reflecting on different membrane interactions. However, we were only able to measure concentration-dependent changes in this position for interaction with the V2R. Hence, we could neither observe a clear helix-bundle nor C-terminus transferability for these conformational changes and did not further discuss it, as we specifically focused on the transferability of conformational changes. However, we agree with the reviewer that this important site should be discussed. The measured β -arrestin2-F1 signal is now mentioned specifically in line 126 in the results section and additionally in lines 417ff in the context of our discussion of transferability.

The ERK1/2 data raises some technical issues: The authors analyzed ERK1/2 phosphorylation in Control and Δ Q-GRK cells stably overexpressing one of the receptor constructs. However, there could be clonal differences in responses simply due to not using the same background cells to analyse this. At the very least, the authors need to perform rescue experiments where they re-express the GRKs and see if the ERK1/2 responses now switch back to those of cells not deleted for GRK. This experiment is key to be able to exploit the results from this analysis - otherwise the results are not reliable.

We thank the reviewer for this constructive feedback and agree that additional experiments, such as the proposed rescue approaches, would significantly strengthen our study's ERK1/2 signaling section. To assess whether our measurement systems are suitable for this assessment, we performed additional western blot analysis for the V2R, including conditions that feature the stable overexpression of GRK2 or GRK6 in Δ Q-GRK cells (Figure I).

This limited experimental series suggests that the expression of GRKs has a negative effect on V2R-induced ERK1/2 phosphorylation, as the highest levels were measured in the absence of GRKs (Δ Q). Overexpression of GRK2 or GRK6 in Δ Q-GRK cells accordingly reduces the obtained phosphorylated ERK1/2 signals to a similar level as seen in Control cells that feature the endogenous expression of GRK2/3/5/6.

Experiments with a similar scope for the b2AR have already been performed in Nobles et al. 2011 (<https://doi.org/10.1126/Scisignal.2001707>), via knockdown of individual GRK isoforms. Although we agree that the direct comparison between the receptors is interesting, the systematic comparison of all receptor

variants over the different time points and GRK conditions is out of the scope of our study.

Figure 1. **a** shows four biological replicates of limited time-course experiments, utilizing four cell lines stably overexpressing HA-V2R, alongside an endogenous complement of GRKs (Control), the quadruple GRK2/3/5/6 knockout (Δ Q), as well as the stable overexpression of GRK2-YFP in Δ Q and GRK6-YFP in Δ Q. Cells were stimulated with 100 nM AVP for 0, 2 and 5 minutes and lysed subsequently. The lysates were then used to visualize phosphorylated ERK1/2 and total ERK1/2 via SDS-PAGE and Western Blot. **b** further expands this analysis for the first biological replicate, additionally visualizing Vinculin, GRK2, GRK6 and HA-V2R expression. As GRK2 and GRK6 were C-terminally fused to YFP to facilitate FACS sorting of similar expression levels, they exhibit increased molecular weight in the blots compared to the endogenous GRKs in Control cells.

However, we also agree with the overall assessment of the reviewers regarding the over-interpretation of our pERK1/2 data. Here, we suspect that GRKs and arrestins have multiple functions that might lead to counterintuitive results upon overexpression of these proteins, as desensitization of the receptor and scaffolding of MAPKs would influence our measured ERK1/2 signaling responses and unfortunately, our setup does not allow for a clear attribution of these effects. Here, we would like to focus on the clear, receptor core-dependent time courses of ERK1/2 activation, keeping our interpretation of arrestin- and GRK-dependent effects to a minimum, as our study mostly focuses on the assessment of β -arrestin2 conformational changes in response to receptor activation. While the influence of GRKs and arrestins on GPCR-mediated ERK1/2 responses still remain underexplored in living cells, future studies need to assess these in further detail, as more of these analyses would exceed the scope of the study at hand.

The ERK1/2 results with Barr overexpression are overinterpreted: Overexpression of barr1 has no significant effect on ERK phosphorylation and that of barr2 only has an effect in chimeric receptors, not in normal ones. The authors need to state this and tone down their assumptions in terms of the role of barr1/2 in ERK1/2 signaling, or perform alternative experiments (with barr1/2 KDs) that support their claims.

We completely agree with the reviewer's verdict regarding our interpretation of ERK1/2 experiments. Following our description of performed experiments, we thoroughly revised the ERK1/2 signaling section of the manuscript ranging from line 215 to line 231. The interpretation of our results has been toned down, as now we are mostly focusing on the distinct time-courses of ERK1/2 phosphorylation mediated by the probed receptor variants. Most interpretations regarding GRK- and β -arrestin-dependency have been removed.

For the endocytic localisation data, the kinetics are not taken into account as only a time point has been imaged. Time lapse microscopy experiments would be more suitable to look at this.

We thank the referee for their suggestion. To address this, we recorded time-lapse confocal microscopy movies in Control and Δ Q-GRK cells for all four receptor constructs and β -arrestin2. The cells were stimulated over 15 min and images were taken every 30 sec. The colocalization data over time was analyzed using Squassh and SquasshAnalyst, analogously to the basal and stimulated images. The results from this data set are now presented in a new supplementary figure (Suppl. Fig. 8). Unfortunately, due to the temporal resolution and the limited sample size inherent to the labor-intensive confocal microscopy setup, we cannot draw detailed conclusions about the internalization kinetics. For future experiments, we are developing stable cell lines, expressing SNAP-tagged GRK constructs, as the YFP-labeled ones would disturb our measurements with β -arrestin-YFP and with transient overexpression of unlabeled GRK constructs, the uncertainty of GRK expression is too pronounced for the rather small sample sizes our confocal microscopy set-up allows. Nevertheless, in Control cells featuring the endogenous GRK complement, we observed a time-dependent increase in receptor co-localization with Rab5, the early endosome marker utilized in this study. To complement the quantification of the confocal imaging, we also prepared representative time-lapse movies in Control cells to visualize the observed cellular reaction over time for each receptor. These movies can be accessed as online material.

Additionally, in Fig 5 - how does barr2 reach the endosome without b2V2 for GRK6 expressing cells? Please give an explanation for these puzzling results or at least an hypothesis of what you think it is going on.

We agree with the reviewer that this detail in the presented datasets is counter intuitive. However, while receptor delivery to early endosomes seems to be massively reduced in the named condition, it is not zero. Our co-localization data shows that approximately 10% of the b2V2-CFP receptor signal co-localizes with Rab5-mCherry, indicating that internalization is not completely halted. Combined with recent reports that β -arrestin2 was shown to be able to retain active conformational states even after dissociation from receptor (Nuber

et al. 2016, <https://doi.org/10.1038/nature17198>; Eichel et al. 2018, <https://doi.org/10.1038/s41586-018-0079-1>) this could explain the accumulation of β -arrestin2, even in subcellular compartments that are not preferably targeted by receptor trafficking.

Currently, several studies focus on localized signaling via the subcellular localization of GPCR interaction partners such as GRKs (Gardner et al. 2024, <https://doi.org/10.1126/scisignal.add9139>) or β -arrestins (Pham et al. 2024, <https://doi.org/10.1101/2024.09.24.614742>), how this “pool” of β -arrestins arrives at endosomes, however, is unclear. As this study mostly focusses on GPCR-induced conformational changes of β -arrestin2, a clear disambiguation of these effects would exceed the scope of our current work, nonetheless we wanted to show the data as it is to display the complete comparison for all conditions.

Reviewer #2 (Remarks to the Author):

Edda Matthees et al. "Relating GPCR domains with functionality: receptor helix-bundle and C-terminus differentially influence GRK-specific functions and beta-arrestin-mediated regulation"

In their interesting manuscript, the Hoffmann team presents a plethora of data on various downstream effects of two prototypical GPCRs, the beta2-adrenergic (beta2AR) and the V2 vasopressin (V2R) receptors, plus two constructs where the C-termini have been swapped. A whole set of downstream effects is measured: GRK-dependent phosphorylation of proximal and distal C-terminal domains (including GRK subtype-selectivity), beta-arrestin recruitment (including beta-arrestin1 vs. -2 selectivity), ERK1/2 activation, and receptor internalization (including subcellular localization and co-localization with rab5).

This results in a large array of data and possible correlations, and I have to confess that I had to read the manuscript several times to understand the main conclusions (and apologies that I needed the Christmas break to get this done!).

We thank the referee for taking the time to review our manuscript and the constructive feedback.

The key points that the authors aim to address is, whether specific downstream effects are mediated more via the receptor core (=helix-bundle) or its C-terminus. To be able to address this, they rely on two receptors considered as prototypes for tight (=class B, V2R) or transient (class A, β 2AR) β -arrestin binding. They further make use of the well-established observation that beta-arrestin1 generally forms less tight interactions with receptors than beta-arrestin2.

The manuscript contains a plethora of carefully obtained and well-documented data, and quite obviously there are limits to what one could wish. For example, having only one data point per log unit may seem little for curve fitting purposes – but if we consider the effort of obtaining such complex data and given the many downstream pathways that are investigated, I believe that the authors have made every reasonable effort.

We thank the reviewer for their positive assessment of our work.

However, before going into any details, I believe that the manuscript would significantly benefit from three changes:

- The authors seem to shy away a little from precise conclusions and remain vague both in the abstract and in the discussion (particularly towards the end of the discussion). Focusing on the most significant findings (rather than describing everything one sees in the data – particularly in the results section, but also in the discussion) and drawing more explicit conclusions would make this manuscript more concise and, in my view, more interesting.

We also agree with the referee that the wealth of data discussed in this study led to certain paragraphs being convoluted and cumbersome to read. Hence, the presentation of our results in the abstract has been changed to more accurately focus on our main conclusions: “Focusing on prototypical class A (b2AR) and B (V2R) receptors and their chimeras (b2V2/V2b2), we show that most N-domain β -arrestin conformational changes are mediated by receptor C-terminus-interactions, while C-domain conformations respond to the helix-bundle or an individual combination of interaction interfaces. Moreover, we demonstrate that ERK1/2 signaling responses are governed by the GPCR helix-bundle, while β -arrestin co-internalization depends on the receptor C-terminus. However, receptor internalization is controlled via the overall GPCR configuration.” All changes are marked in yellow in the manuscript version with tracked changes to facilitate the direct comparison.

Furthermore, we toned down our interpretation of ERK1/2 experiments in the results section and removed several paragraphs from the lengthy discussion. Here, we hope that those text changes improve readability and specifically highlight our most important findings.

- The use of just two receptors plus two C-tail swapped variants obviously limits the generalizability of the findings and conclusions. Contrasting this with my wishes expressed above, it is certainly difficult, but in my view possible, to be somewhat more careful in postulating general principles and yet, at the same time, draw more explicit conclusions from the experiments.

We agree with the reviewer that our sample size of two probed receptors and variants might be limited, but also the vast data sets that we collected for these conditions complicate the clear interpretation of some results. Specifically, as GPCR regulating proteins, such as GRKs and β -arrestins likely fulfill multiple different functions, there are several dimensions to interpret obtained results. Here, we wanted to provide a tangible interpretation of receptor domain-dependent effects via our analysis of transferability (Figure 6), with a focus on β -arrestin2 conformational changes.

However, we reworked the manuscript text and tried to specifically emphasize our key results without over-generalizing our findings. Here, we thoroughly

revised the abstract to clarify the interpretation of obtained results. Additionally, the ERK1/2 signaling section in the discussion has been changed to avoid the postulation of general principles (lines 460ff) and the discussion overall has been shortened to emphasize the most important points and avoid repetitive sections.

- The authors mention the publications on GPCR/beta-arrestin complex structures and they provide structural images in many of their figures. However, I see little link between these structures and the specific mechanisms proposed in this manuscript. I would suggest strengthening such links – if at all possible

We agree with the reviewer that the molecular interpretation of our data represents an important link between pharmacological analysis and current structural approaches. However, it is a limitation of our biosensor system that conformational changes measured for individual FIAsh positions cannot be used for further analysis, such as approximation of change in distance between the energy donor and acceptor, as these represent ensemble measurements that yield average conformational change signals of all subcellular pools of β -arrestins. Structures in this manuscript are specifically used as visual aids, enabling snapshot views of how these complexes could look like.

Single molecule approaches, as featured in Asher et al. 2022 (<https://doi.org/10.1016/j.cell.2022.03.042>) could bridge this gap, however, they bear their own limitations, as they neglect the influence of the cellular architecture via the use of purified proteins.

Hence, we try to focus on the interpretation of collective datasets in the form of full conformational fingerprints. Here it is permissible to interpret global trends, such as N-domain versus C-domain conformational changes, or receptor C-terminus interactions versus membrane interactions. In accordance with this reviewer comment, an additional interpretation of β -arrestin2 interactions with the membrane has been added (lines 418-423) to include another level of structural interpretation of our biosensor data, specifically highlighting F1 conformational changes.

Specific points:

1. In the abstract, I would suggest to be much more explicit about key differences observed, and to eliminate statements such as “some processes – while others...”. This would allow the reader to start reading with the authors’ main conclusions in mind.

The abstract has been thoroughly revised to follow the reviewers’ suggestions.

2. In the kinetic experiments (Fig. 1) it is not clear at what time point agonist was added (at t=0?) and why the exponential part of the calculated curves begins at

very defined time points. In some traces, there are no data near these time points, which makes curve fitting a little arbitrary.

Time point 0 has been calculated as the average time needed to facilitate ligand addition via electric multichannel pipettes, considering all analyzed biological and technical replicates. The data was fitted using the pharmechanics plugin for prism (Hoare et al. 2020, <https://doi.org/10.1038/s41598-020-67844-3>) with X0 being restricted to > 0 min. This description has now been added to the legend of Figure 1.

3. In Fig. 2 I would suggest concentrating essentially on the radar plots, which summarize the large number of curves depicted in panels c-r. These radar plots seem to show that β -arrestin conformational changes are mostly dictated by the receptor's C-terminus (V2R better than β 2AR). Exceptions appear to be the agonist-dependent changes that are also seen with β 2AR (notably F2, F3, F10).

We agree that these are vast datasets and the radar plots specifically are the most important display items to describe GRK- and receptor-dependent arrestin conformational change fingerprints. However, we additionally analyzed other aspects of GRK-dependent receptor regulation (Control versus Δ Q), to complete this picture via GRK-specific C-terminal receptor phosphorylation (a-j), β -arrestin recruitment (k-n) and GRK-dependent conformational changes (o-r, F5 conformational changes are shown to give a fair impression of the data that the radar charts are based on). The radar plots do not summarize panels c-r, as these show 3 different assays/aspects as listed above. To emphasize the structure of the figure, we included sub-headings that specifically indicate the performed assays on top of the respective panels. With the aim of making the individual panels less crowded, we updated the figure to include one general legend for each subsection, instead of each individual panel.

Otherwise, we agree with the reviewer and hope that our interpretation comprehensively describes the differences in arrestin conformational states induced via the different receptor variants.

4. The idea that the pattern of ERK activation (short vs. long lived) is mediated via the receptor core and is, hence G-protein-mediated, is an attractive hypothesis (Fig. 3). However, it might be appropriate to accommodate also the thoughts of those researchers who disagree with that notion. It might also be good to contrast ref 15 with those publications that argue for a major (and perhaps G-protein-independent) role of β -arrestins in this process.

We thank the referee for pointing this out. Our introduction (lines 42ff) features a discussion of ERK1/2 signaling responses and the proposed impact of β -arrestins on this signaling. However, we generally toned

down our interpretation of GRK- and β -arrestin-mediated ERK1/2 signaling responses, as it is now evident that our analyses do not allow for a clear attribution of positive or negative effects. Furthermore, we believe that adding additional experiments to this already vast study would only convolute the manuscript at hand – further research has to be conducted to uncover this complex relationship between GPCR regulating proteins, such as GRKs and arrestins and their influence on GPCR-induced MAPK signaling.

5. It seems appropriate, given the demanding experiments, to use only one example each of GRK2,3 and GRK5,6 in experiments on GRK specificity (Fig. 4) – but as in the case for the only two receptors (see above) it might be appropriate to add a word of caution that differences to the subtypes not tested might be possible. In Fig. 4, I would again concentrate on describing the radar plots. A specific interesting effect might be the observation that F4 responds very differently to the two receptor tails.

We agree with the reviewer that the influence of individual GRK isoforms might differ, even though it seems that GRK2 and 3, as well as GRK5 and 6 induce mostly overlapping functions. As suggested, a cautious statement has been added in lines 243ff.

Phosphorylation data are in our opinion key here, as they show that the pattern of conformational changes in β -arrestin are not directly resulting from the degree of measurable phosphorylation, as generally the detected β -arrestin conformational changes in presence of GRK2 were more pronounced than in presence of GRK6.

To improve the didactic flow of our figures, we included sub-headings that specifically indicate the performed assays on top of the respective panels. We hope that this change adds an additional visual structure to clarify that these experiments do not directly target β -arrestins but focus on the receptors, in contrast to the radar plots below.

Furthermore, we agree with the reviewer that observed changes in β -arrestin N-domain at the F4 position are indeed interesting. We were also able to record these differences in Control cells in Figure 2 and it seems that these findings are consistent with the phosphorylation sensing binding sites in β -arrestin N-domain, which likely explains the different responses.

This is already discussed in the manuscript, e.g. in line 403ff: “Comparing wild type and chimeric receptor-induced β -arrestin conformational changes using the transferability coefficient, we noticed a consistent trend of N-domain conformational changes (F2, F4, F5) being C-terminus transferable under endogenous GRK expression (Fig. 6a-d). We showed that these are predominantly GRK-dependent (Fig. 2s-v). This phenotype was particularly pronounced for position F4. Notably, in the complex crystal structure of the V2R with β -arrestin, the phosphorylated C-

terminus closely interacts with sites adjacent to the F4 position. Given that the β -arrestin N-domain serves as the primary interaction site for phosphorylated receptor domains, it is not surprising that conformational changes in this region are strongly influenced by the specific GPCR C-terminus.”

6. The fact that endosomal localization is so strikingly specific for the receptor C-tail (Fig. 5) is very interesting. Can the authors suggest a mechanistic explanation? And how does this fit with much less pronounced specificity for β -arrestin conformational changes.

Based on our data, we expect that the “information” about class A/class B co-internalization with β -arrestin might be encoded within the receptor C-terminus (possibly influencing the affinity of β -arrestins to receptor), whereas overall β -arrestin conformational changes, specifically in different regions of β -arrestins, depend on multiple additional factors (such as the receptor core, or other interacting proteins within a signaling complex – see e.g. review Haider et al. 2023, <https://doi.org/10.1002/bies.202300053>). According to the reviewer’s suggestion, a mechanistic explanation has been added to the discussion (lines 568ff).

7. I like the use of the transferability concept in Fig.6 to assign specific functions to core vs. C-tail. As suggested above – can this be interpreted structurally?

We thank the reviewer for their positive assessment and agree that the experiments included in the manuscript at hand reveal striking insights into the structure-function relationship of distinct receptor- β -arrestin complexes. Our interpretation however has to be in accordance with the structural resolution of performed experiments – hence, we specifically focus on interpreting the influence of individual receptor domains (C-terminus vs receptor core), with overall β -arrestin conformational states and a limited selection of downstream functions. Here, clear examples feature the time-dependent mediation of ERK1/2 phosphorylation, as receptor core-transferable (as a direct result of G protein signaling), and β -arrestin endosomal delivery that shows to be GRK-dependent and C-terminus-transferable (as this domain serves as the main substrate for phosphorylation). Effects for which we cannot assign clear transferability specifically indicate a more complex interplay between structural domains.

Minor point:

The authors use the terms arrestin and beta-arrestin interchangeably and most of the times it is not clear to me why they use which expression. I know there is some confusion (and controversy) about the right terminology - but it would

make sense to stay within one terminology. In this case, since all data are on beta-arrestins, I would suggest to call them beta-arrestins.

Specifically, in the first paragraphs of the introduction, we refer to “arrestins” as all four human isoforms (arrestin1, β -arrestin1, β -arrestin2 and arrestin4). However, we agree with the reviewer that indeed, we mostly discuss β -arrestins. Hence, where accurate, the name “arrestin” has been removed and replaced with “ β -arrestin”.

Reviewer #3 (Remarks to the Author):

This paper examines the distinct roles that the C-tail and the central GPCR helical bundles play in driving beta-arrestin conformational changes following activation by canonical class A GPCR, i.e. the β 2-adrenergic receptor (β 2AR with subscripted 2) and class B GPCRs, i.e. the vasopressin V2 receptor. They use established approaches to assess β -arrestin recruitment and β -arrestin conformation and provide an interesting analytical framework in which to assess the roles of different phosphorylation sites in the receptor C-tails and distinct conformational effects on the N- and C-terminal domains of β -arrestin.

We thank the reviewer for the concise summary of our work and positive feedback.

Comments

In general the experiments are well designed, well conducted and well analyzed. The use of the different receptor chimeras and the CRISPR KO lines is a strength. Some things the authors could address include:

Major:

1) I'm not sure the magnitude of BRET changes (as shown in Figure 1e,f) are meaningful as such. Are the levels of biosensor expression similar? I assume since these are intramolecular BRET constructs there is no issue of stoichiometry but it would have been a useful control to include in supplementary data. Different receptor expression could also dramatically contribute to this change of magnitude.

We agree that the magnitude of measured signals is only meaningful if compared to the entire dataset in form of a “fingerprint”. According to the reviewer’s suggestion, one sentence highlighting receptor-specific differences in signal amplitude has been removed (line 105f).

2) Can the authors explain in better detail the odd results in Figure 2l, n? Could it be more nuanced than a simple effect of different helical cores? Moreover, L199: “we were also able to monitor a residual GRK-independent β -arrestin2 recruitment to all receptor variants, except the V2b2...” How can the authors reconcile the absence of β arr2

recruitment by V2b2 in ΔQ -GRK condition (Figure 2i) with the change of β arr2 conformational change fingerprint reported in Figure 2t ?

We agree with the reviewer that the absence of measurable β -arrestin2 recruitment to the V2b2 in ΔQ is unexpected, specifically as we were able to show that stable interactions between the V2R and β -arrestin2 occur even in the absence of GRKs. This results in our interpretation of β -arrestin recruitment phenotypes in ΔQ not being clearly core transferable, even in the assumed absence of phosphorylation. This might indicate a more complex structural relationship of individual arrestin-interacting GPCR domains that potentially results in a different geometry of formed complexes. This is specifically relevant if we consider the available complex structures of β -arrestin1 with the β 1 adrenergic receptor (PDB: 6TKO) and the V2R (PDB: 7R0C), which indicate that β -arrestin1 binds to these receptors with an approximately 30° rotation (Underwood et al. 2024, <https://doi.org/10.1111/bph.16331>). With this, it could be that even if the arrestin finger loop region is inserted into the intracellular cavity of different receptors distinct complex configurations might occur. Here, the C-terminus could still be an additional stabilizing factor, even in the absence of GRK-mediated phosphorylation.

Hence, it is also possible that miniscule β -arrestin2 recruitment does in fact occur, yet either the interaction might not be stable enough or the resulting geometry of the complex does not allow for monitoring via our BRET method.

3) Figure 3: For ERK activation, the authors normalized the data as fold change compared to the untreated cells (n=1), then established some comparisons between ctrl vs KO-GRK. However, there is no certainty the basal phosphorylation of ERK is equivalent for every condition (difference in constitutive GPCR activity for instance) Hence, does KO GRK really decrease ERK activation, or could it augment receptor basal activity (especially considering that overexpressing GRK increased basal phosphorylation in Fig 4)? Does β arr overexpression really increase ERK activation, or in fact reduce receptor constitutive activity? Immunoblotting is a semi-quantitative method, as such, without comparing the basal levels between conditions on a same gel, any mechanistic claims the authors make about β arr involvement in ERK activation appears over-interpretative (such as I51 “Moreover, we demonstrate that some β -arrestin-supported”, I270 “ β -arrestin-supported mechanism”...). Hence, the limitation in I554 “possibly also influencing initial GRK and β arrestin interaction” should be expanded in the ERK section, and comments about β arr involvement with ERK ... maybe be rephrased with consideration.

We agree with the reviewer that some of our experiments evaluating ERK1/2 activation might have been over-interpreted in our original draft. Hence, we overhauled the ERK1/2 signaling section and now mostly focus on the interpretation of different time courses, that are evidently induced by V2R and b2AR helical cores.

However, untreated cells used for these quantifications were not $n=1$. On the contrary, untreated sample was included on every experimental day, visualized on the same gel and Western blot, with data being individually normalized to this respective untreated value.

To assess possible differences in total ERK1/2 and phosphorylated ERK1/2 signal between different cell lines, we performed additional Western blot analyses with stored (-20°C) lysates for receptors featuring the b2AR core (b2AR, b2V2) on one large gel per receptor (Figure II). Furthermore, we performed additional experiments ($n=3$) with new lysates on one large gel for V2R and V2b2, respectively (Figure III). Here, we would like to argue that the comparison between cell lines seems permissible, as no striking differences were observed between basal levels are measurable between ΔQ and Control cells, with and without β -arrestin overexpression.

Figure II. Lysates ($n=3$) of cellular conditions featuring b2AR receptor cores, for empty vector transfection (EV) and cells overexpressing either β -arrestin1 or 2, at time points 0 (untreated) and after two minutes of stimulation with $1 \mu\text{M}$ isoproterenol, visualized on a large gel, respectively for each receptor. The three independent ns per condition are shown right next to each other.

Figure III. Lysates ($n=3$) of cellular conditions featuring V2R receptor cores, for empty vector transfection (EV) and cells overexpressing either β -arrestin1 or 2, at time points 0 (untreated) and after five minutes of stimulation with 100 nM AVP, visualized on a large gel, respectively for each receptor. The three independent ns per condition are shown right next to each other.

4) What does the grey scale indicate in the heatmap?

Similarly to the grey sensor regions plotted in Figure 1i and j, grey rectangles in the heatmap represent “non-responding” biosensor positions. This information has now also been added to the color scheme legend in Figure 4m and n.

5) In Figure 4, could the additional feature beyond receptor phosphorylation per se be based in the presence or interaction with distinct pools of Gbetagamma subunits for GRK2 or GRK6?

The reviewer rightfully points out that $G_{\beta\gamma}$ dimers serve as, often underappreciated, adaptors to facilitate receptor desensitization, specifically as they facilitate the recruitment of GRK2 family kinases (GRK2 and GRK3). Matthees et al. 2024 (<https://doi.org/10.1038/s42003-024-06490-1>) shows that these $G_{\beta\gamma}$ -GRK2/3 interactions are essential to enable high-affinity β -arrestin interactions, while GRK5 and 6, however, are not described to interact with $G_{\beta\gamma}$. Here we agree with the reviewer that these effects are very interesting and definitely an integral part of the receptor regulation cascade, yet we fear that including $G_{\beta\gamma}$ -specific conditions in this manuscript would further convolute the interpretation of obtained results.

However, there are also other reasons that could explain the different levels of receptor phosphorylation induced by either GRK2 or GRK6. Stable cell lines utilized for our bead-based receptor phosphorylation assays were sorted for similar GRK levels using conjugated YFP, however, GRK2 as a cytosolic kinase, is evenly distributed within the three-dimensional volume of cells, while membrane-localized GRK6 concentrates at cell surface, possibly resulting in different local concentrations

6) Do distinct ligands for each receptor drive distinct effects? This might be an interesting addition to the story.

We thank the reviewer for this insightful comment. Indeed, we expect that different ligands, provided they induce distinct active receptor conformational and/or phosphorylation states, would influence the resulting conformation of bound β -arrestins and subsequent β -arrestin-mediated functions. As shown already in Reiner et al. 2010 (<https://doi.org/10.1074/jbc.M110.175604>), focusing on different agonists for the b2AR, different ligands are capable to influence these receptor activation metrics and by inducing distinct receptor conformational changes, they are expected to drive different effects. However, including just one additional ligand for b2AR and V2R receptor cores, respectively, would double our already vast datasets, possibly complicating the interpretation of gained results. Future studies have to focus on these ligand-specific effects, to uncover how different agonists fine-tune arrestin-mediated functions.

7) Comparing transferability coefficients between Figure 6B and Supp Figure 11, we see the pattern can vary drastically depending on GRK expression. This means there

is no universal transferability pattern regarding β -arrestin2 conformation... As such, authors should correct their conclusion l558: "By focusing on two of the most prominent class A and B receptors, β 2AR and V2R, as well as their chimeras, we have discerned that N-terminal β -arrestin2 conformational change patterns are primarily C-terminus transferable under endogenous GRK expression IN HEK 293 cells".

We thank the reviewer for this valuable suggestion. This information has been added according to the reviewers suggestion.

Minor:

1) Although β -arrestin is correctly written, the authors refer to b2AR rather than β 2AR.

We agree with the reviewer that β 2AR would be the preferred abbreviation according to the common usage. However, after discussion with the editor, we collectively decided that "b2AR" is an appropriate abbreviation, specifically as to not confuse the receptor with β -arrestin2. Furthermore, keeping the abbreviation as it is would also allow us to be consistent with our own work, previously published in Nat Commun (Drube et al. 2022, <https://doi.org/10.1038/s41467-022-28152-8>). Nonetheless, we have added this statement also to the method section of the manuscript at hand to provide clear information regarding the naming convention.

2) Did they make 8 new lines for Figure 4 or sixteen? Is this because some had β -arrestin1 or 2 as well?

This has been rephrased in lines 249, now clearly stating the number of stable transfectants. The total number of stable cell lines created for this study was sixteen – four different kinase conditions for each investigated receptor variant, the investigation of effects of β -arrestin1 or 2 was performed in transient overexpression experiments.

3) L117 l117: "Using a refined set of β -arrestin conformational biosensors, we recently uncovered arrestin conformational fingerprints that are induced by ligand-activation of the presumably unphosphorylated parathyroid hormone 1 receptor (PTH1R), in the absence of GRKs25. This finding confirms that both binding interfaces, namely the GPCR transmembrane helix-bundle and C-terminus, influence the resulting active arrestin conformation" : The first sentence is an experimental approach, not a finding. Please insert the main result of (25) that is supporting the second sentence.

This has been rephrased and the word "finding" has been omitted. Here, we only want to point out that already previously we were able to detect GRK-independent β -arrestin conformational changes, which suggests that the PTH1R is able to activate β -arrestins even in the absence of GRKs.

4) I506: Regarding ERK: “Since the overall kinetics of ERK activation did not change in our experiments, we cannot conclude that the GRK-mediated β -arrestin interaction is solely responsible for the prolonged phase. However, other GRK-specific functions may also contribute to this phenomenon, such as internalization.”, GRK-specific appears paradoxical since there is no kinetic difference between control and KO-GRK.

Overall, we toned down our interpretation of ERK1/2 phosphorylation western blot analyses. The sentence now only clarifies that there might be different ways GRKs could (positively or negatively) influence MAPK signaling responses. Further research needs to be done to clearly uncover those interactions, yet this is not the main focus of the manuscript at hand.

Reviewer #4 (Remarks to the Author):

We thank the referee for taking the time to review our manuscript and to integrate his/her constructive feedback into the comments.